J Physiol 601.16 (2023) pp 3647–3665

# MitoQ as an antenatal antioxidant treatment improves markers of lung maturation in healthy and hypoxic pregnancy

Mitchell C. Lock[1] [iD], Kimberley J. Botting[2] [iD], Beth J. Allison[2] [iD], Youguo Niu[2] [iD], Sage G. Ford[2], Michael P. Murphy[3], Sandra Orgeig[4], Dino A. Giussani[2] [iD] and Janna L. Morrison[1] [iD]

[1] *Early Origins of Adult Health Research Group, Health and Biomedical Innovation, UniSA: Clinical and Health Science, University of South Australia, Adelaide, South Australia, Australia*

[2] *Department of Physiology, Development & Neuroscience, University of Cambridge, Cambridge, UK*

[3] *MRC Mitochondrial Biology Unit, University of Cambridge, Cambridge, UK*

[4] *UniSA: Clinical and Health Science, University of South Australia, Adelaide, South Australia, Australia*

Handling Editors: Laura Bennet & Christopher Lear

The peer review history is available in the Supporting Information section of this article (https://doi.org/10.1113/JP284786#support-information-section).

The Journal of Physiology

**Abstract** Chronic fetal hypoxaemia is a common pregnancy complication that increases the risk of infants experiencing respiratory complications at birth. In turn, chronic fetal hypoxaemia promotes oxidative stress, and maternal antioxidant therapy in animal models of hypoxic pregnancy has proven to be protective with regards to fetal growth and cardiovascular development. However, whether antenatal antioxidant therapy confers any benefit on lung development in complicated

D. A. Giussani and J. L. Morrison contributed equally to this work.

pregnancies has not yet been investigated. Here, we tested the hypothesis that maternal antenatal treatment with MitoQ will protect the developing lung in hypoxic pregnancy in sheep, a species with similar fetal lung developmental milestones as humans. Maternal treatment with MitoQ during late gestation promoted fetal pulmonary surfactant maturation and an increase in the expression of lung mitochondrial complexes III and V independent of oxygenation. Maternal treatment with MitoQ in hypoxic pregnancy also increased the expression of genes regulating liquid reabsorption in the fetal lung. These data support the hypothesis tested and suggest that MitoQ as an antenatal targeted antioxidant treatment may improve lung maturation in the late gestation fetus.

(Received 3 April 2023; accepted after revision 4 July 2023; first published online 19 July 2023)

**Corresponding author** J. L. Morrison: Head, Early Origins of Adult Health Research Group, Health and Biomedical Innovation, UniSA: Clinical and Health Science, University of South Australia, GPO Box 2471, Adelaide, South Australia, Australia 5001. Email: Janna.Morrison@unisa.edu.au

**Abstract figure legend** Summary of molecular changes within the fetal lung as a result of MitoQ treatment. Changes in expression of pathways are indicated by direction of arrows; no change in expression is represented by purple sideways double-sided arrows. A main effect of MitoQ is represented by a green arrow. An interaction of hypoxia and MitoQ is indicated by yellow arrows. MitoQ treatment increased the expression of mitochondrial complex III and ATP synthase (Complex V) and expression of surfactant proteins B and C. There was an interaction of MitoQ with hypoxia resulting in increased expression of sodium transporter SCNN1A and surfactant protein D within the fetal lung.

## Key points

- Chronic fetal hypoxaemia promotes oxidative stress, and maternal antioxidant therapy in hypoxic pregnancy has proven to be protective with regards to fetal growth and cardiovascular development. MitoQ is a targeted antioxidant that uses the cell and the mitochondrial membrane potential to accumulate within the mitochondria.
- Treatment of healthy or hypoxic pregnancy with MitoQ, increases the expression of key molecules involved in surfactant maturation, lung liquid reabsorption and in mitochondrial proteins driving ATP synthesis in the fetal sheep lung.
- There were no detrimental effects of MitoQ treatment alone on the molecular components measured in the present study, suggesting that maternal antioxidant treatment has no effect on other components of normal maturation of the surfactant system.

## Introduction

A reduction in fetal oxygenation or chronic fetal hypoxaemia is one of the most common outcomes in complicated pregnancy (Giussani, 2016). Chronic fetal hypoxaemia may result from maternal, placental or fetal complications. These may include pregnancy at high altitude, maternal smoking, preeclampsia, placental insufficiency or umbilical cord compression (Hutter et al., 2010). A major risk factor for hypoxic pregnancies is the development of respiratory distress syndrome (RDS) in the neonate, due to alterations in the development of the surfactant and antioxidant systems within the fetal lungs during late gestation (Avery & Mead, 1959; McGillick et al., 2016).

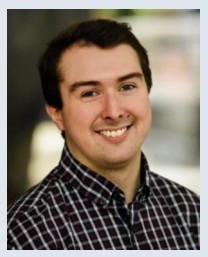

**Mitchell C. Lock** is an early career researcher within the Early Origins of Adult Health Research Group at the University of South Australia. Mitchell's research focuses on understanding the molecular mechanisms by which sub-optimal *in utero* conditions such as fetal hypoxaemia can influence lung maturation and cardiac development and the programming of adult cardiovascular disease. More recently he has been focused on investigating the therapeutic potential of antenatal antioxidants and how their use in hypoxic pregnancy may provide protection for the lung and heart both in early postnatal life and adulthood.

Antioxidants play a vital role in protecting the developing fetal lung from oxidative stress induced by exposure to changing levels of oxygenation (Berkelhamer & Farrow, 2014). The antioxidant system in the lung matures across gestation (Frank & Sosenko, 1987), with an increased expression and activity of antioxidant enzymes, including glutathione peroxidase (GPX), superoxide dismutase (SOD) and catalase (CAT). Complicated pregnancies are often associated with changes in pro- and antioxidant profiles during fetal development due to increased reactive oxygen species (ROS). Alterations in the development of the lung surfactant and antioxidant systems contribute to a poor transition from the fetal to the postnatal period, and the establishment of RDS (Avery & Mead, 1959; Frank & Sosenko, 1987). For instance, oxidised surfactant proteins (SPs) have reduced surface tension, exacerbating respiratory distress in the newborn of complicated pregnancy (Hallman, 1997; Holm et al., 1988; Nogee et al., 1991; Putman et al., 1997; Saugstad, 1998).

To combat oxidative stress in at-risk pregnancies antenatal antioxidant treatment has gained increased traction in recent years as potential therapy (for review, see Giussani, 2021). Previous studies report that maternal antioxidant treatment such as with vitamin C, melatonin or allopurinol can protect against programmed cardiovascular disease in offspring of an hypoxic pregnancy in sheep (Botting et al., 2020; Giussani et al., 2012, 2021) and rats (Botting et al., 2020; Hansell et al., 2022; Kane et al., 2013; Niu et al., 2018; Smith et al., 2022). Importantly, vitamin C treatment in the last third of pregnancy in healthy sheep acts at the molecular level to increase the expression of genes that are important for fetal lung maturation (McGillick et al., 2021). However, conventional antioxidants, such as vitamin C, are only effective at very high doses, incompatible with human translation (Botting et al., 2020). Since mitochondria are a major site of ROS production, mitochondria-targeted antioxidant therapy may provide a more focused therapeutic strategy. MitoQ is a promising candidate and is a bioactive ubiquinone (Coenzyme $Q_{1n}$) conjugated to a lipophilic triphenylphosphonium cation, which promotes bioaccumulation 100- to 500-fold within the mitochondria in fetal tissues (Botting et al., 2020; Nuzzo et al., 2018; Spiroski et al., 2021). This highly targeted approach has demonstrated effectiveness in combating several pathologies resulting from oxidative stress in a range of studies in rodents, sheep and in human clinical trials (Botting et al., 2020; Fortner et al., 2020; Gane et al., 2010; Pinho et al., 2020; Snow et al., 2010). However, whether antenatal MitoQ therapy may confer any benefit on lung development in complicated pregnancies has not yet been investigated. Here, we tested the hypothesis that maternal treatment with MitoQ in late gestation will improve markers of maturation in the developing lung in both normal and hypoxic pregnancy in sheep, a species with similar fetal lung developmental milestones as humans (Lock et al., 2013; Morrison et al., 2018).

## Methods

### Ethical approval

Experimental protocols for animal research were performed under the UK Animals (Scientific Procedures) Act 1986 and were approved by the Ethical Review Committee of the University of Cambridge under Home Office Project Licence PL70/7645 and PL80/2232. Experiments were designed and reported with reference to the ARRIVE guidelines (Kilkenny et al., 2010). The experiments comply with the policies and regulations of *The Journal of Physiology* (Grundy, 2015). In total, 36 Welsh mountain ewes carrying singleton pregnancies were used in this study. All animal studies were performed at the University of Cambridge (UK) and molecular analyses were performed at the University of South Australia (Australia).

### Surgical procedure

Pregnant ewes carrying a singleton pregnancy (determined by ultrasound scan at 80 days of gestation; term, 145 days) underwent surgery under general anaesthesia using aseptic conditions at $100 \pm 1$ days of gestation. Anaesthesia was induced by the intravenous infusion of Alfaxan via jugular injection (1.5–2.5 mg kg$^{-1}$, alfaxalone; Jurox Ltd, Worcestershire, UK) and maintained with inhalation of isoflurane (1.5%–2%) in 60:40 $O_2$:$N_2O$. Antibiotics (30 mg kg$^{-1}$ I.M. procaine benzylpenicillin; Depocillin; Intervet UK Ltd, Milton Keynes, UK) and an analgesic (1.4 mg kg$^{-1}$ S.C. carprofen; Rimadyl; Pfizer Ltd, Kent, UK) were administered immediately before the start of surgery. Briefly, laparotomy was performed as previously described (Brain et al., 2015; Gardner et al., 2004). Following a midline abdominal incision and uterotomy, the fetal hind limbs were exposed and fetal sex was determined. If male, fetuses were chosen for this study. Female fetuses were used for other experiments (Brain et al., 2019). The fetus was returned into the intrauterine cavity and the uterine and maternal abdominal incisions were closed in layers. Vascular catheters were then inserted in the maternal femoral artery and femoral vein. Catheters were filled with heparinised saline (100 I.U. ml$^{-1}$ heparin in 0.9% NaCl) and kept inside a plastic pouch sewn onto the maternal skin. Post-surgery, ewes were housed in individual floor pens with a 12 h:12 h light:dark cycle

with *ad libitum* access to hay and water, and 200 g of concentrated pellets. Analgesia was administered for 3 days after surgery (1.4 mg kg$^{-1}$ carprofen, subcutaneously (s.c.); Rimadyl; Pfizer Ltd). Antibiotics were administered intramuscularly to each ewe (30 mg kg$^{-1}$ procaine benzylpenicillin; Depocillin; Intervet UK Ltd, Milton Keynes, UK) daily for 5 days following surgery. Ewes were then transferred to a maintenance diet of pellets appropriate for the nutritional demands of pregnancy (Cambridge ewe diet: 40 g nuts kg$^{-1}$; Manor Farm Feeds Ltd; Oakham, Leicestershire, UK) and 3 g hay kg$^{-1}$ to facilitate the monitoring of food intake. At 105 days of gestation, following 5 days of post-operative recovery, ewes were randomly assigned to one of four treatment groups: normoxic saline (NS; $n = 8$), normoxic MitoQ (NM; $n = 10$), hypoxic saline (HS; $n = 8$) or hypoxic MitoQ (HM; $n = 10$). In treated groups, a daily 5 ml bolus of either saline or MitoQ (6 mg kg$^{-1}$ of MS010 dissolved in saline) was administered every day for 33 days through the indwelling maternal femoral vein catheter, as previously described (Botting et al., 2020).

### Chronic hypoxic exposure

Ewes allocated to hypoxic pregnancy were housed in isobaric hypoxic chambers as previously described (Botting et al., 2020; Brain et al., 2015). Briefly, chambers were filled with hypoxic air (∼11% O$_2$), which was created by mixing compressed air and nitrogen. The mixture of air and nitrogen (10 l s$^{-1}$) was adjusted as required for each ewe in order to achieve a daily maternal $P_{\mathrm{aO_2}}$ in arterial blood of 45−55 mmHg. This degree of maternal hypoxaemia has previously been shown to result in fetal hypoxaemia ($P_{\mathrm{aO_2}}$ 11−12 mmHg) (Allison et al., 2016) comparable to cordocentesis samples in human fetal growth restriction (FGR) (Soothill et al., 1987). Maternal blood gas data following chronic hypoxia in this cohort has been published previously (Botting et al., 2020). The air mixture underwent a minimum of 12 changes per hour. All chambers were equipped with an atomised automatic humidity system (1100-03239 HS-SINF Masalles, Barcelona, Spain) to maintain appropriate humidity (55 ± 10%). Ambient $P_{\mathrm{O_2}}$, $P_{\mathrm{CO_2}}$, humidity and temperature within each chamber were monitored via sensors and recorded continuously throughout the exposure. The chambers were transparent, allowing ewes to see each other. The air mixture was passed via silencers able to reduce noise levels within each hypoxic chamber to 63 dB(A). This value is lower than those necessary to abide by the *Control of Noise at Work Regulations*. This not only complied with animal welfare regulations but also provided a highly tranquil environment for the animal inside each chamber (Allison et al., 2016; Botting et al., 2020; Brain et al., 2019).

### Post-mortem and tissue collection

At 138 days of gestation, ewes were humanely killed via overdose of sodium pentobarbitone (0.4 ml kg$^{-1}$ i.v. Pentoject; Animal Ltd, York, UK). The uterus was opened, and the fetus was removed and weighed. The fetal lungs were dissected and weighed. Lung tissue for molecular analysis was collected from the lower right lobe from the same position in each fetus and immediately frozen in liquid nitrogen for qRT-PCR and western blot analyses.

### Real-time PCR for target genes

All essential information regarding the qRT-PCR procedure is included as per the MIQE guidelines (Bustin et al., 2009). Total RNA was extracted from frozen lung tissue for each fetus using QIAzol Lysis Reagent solution and QIAgen miRNeasy purification columns, as per manufacturer's guidelines (Qiagen, Germany). Total RNA was quantified by spectrophotometric measurements at 260 and 280 nm in a NanoDrop Lite Spectrophotometer (Thermo Fisher Scientific). The 260/280 nm ratio results were less than 2.1 and greater than 1.9 and therefore acceptable for qRT-PCR. cDNA was synthesised using Superscript III First Strand Synthesis System (Invitrogen, USA) using 1 $\mu$g of total RNA, random hexamers, dNTP, DTT and Superscript III in a final volume of 20 $\mu$l, as per the manufacturer's guidelines in a MJ Mini personal thermocycler (Biorad, USA). Controls containing either no RNA transcript or no Superscript III were used to test for reagent contamination and genomic DNA contamination, respectively. The geNorm component of qbaseplus 2.0 software (Biogazelle, Belgium) was used to determine the most stable reference genes from a panel of candidate reference genes (Vandesompele et al., 2002) and the minimum number of reference genes required to calculate a stable normalisation factor, as previously described (Lie et al., 2014; McGillick et al., 2013; Soo et al., 2012). For qRT-PCR data output normalisation, two stable reference genes ACTB and YWHAZ (Table 1) were run in parallel with all target genes, as previously described (Lock et al., 2017). A selection of genes was chosen *a priori* to investigate key pathways involved in lung development. Primers were validated and optimised as previously described (McGillick et al., 2014; McGillick et al., 2021; Orgeig et al., 2010). Relative expression of target genes (Table 1) involved in: SP and phospholipid production (*SFTPA, SFTPB, SFTPC, SFTPD, PCYT1A, ABCA3*), SP regulatory genes (*TTF1, FOXA1, SP1*), glucocorticoid signalling and conversion (*NR3C1, NR3C2, HSD11B1, HSD11B2*), hypoxia signalling (*HIF3A, EGLN3*), nitric oxide synthesis (*INOS, ENOS*), mitochondrial function and transcription factors (*SOD1, SOD2, TFAM, NRF1*), pro and antioxidant

**Table 1. qRT-PCR primer and western blot antibody information**

| Primer name | Primer sequence 5′ to 3′ | Primer conc ($\mu$mol l$^{-1}$) | Accession number |
| --- | --- | --- | --- |
| ACTB | F - CCAAGGCCAACCGTGAGA | 0.45 | U39357 |
| | R - AGCCTGGATGGCCACGT | 0.45 | |
| YWHAZ | F - TGTAGGAGCCCGTAGGTCATCT | 0.45 | AY970970 |
| | R - TTCTCTCTGTATTCTCGAGCCATCT | 0.45 | |
| SFTPA | F - AGCTCCAGGGCACACTCCATG | 0.3 | AF211856 |
| | R - CTCCCACTTCCAGCATGGAC | 0.3 | |
| SFTPB | F - GGGCCCCACATTCTGGTGC | 0.3 | AF107544 |
| | R - TCCTTGGCCATCTTGGTGAGG | 0.3 | |
| SFTPC | F - GCAAAGAGGTCTTGATGGAG | 0.3 | AF076634 |
| | R - CAGGGCTCCTACGATCACC | 0.3 | |
| SFTPD | F - GGCCACAGCCCAGAACAA | 0.3 | AJ133002.1 |
| | R - AAGTACCCTCCTTCCTGGTATCG | 0.3 | |
| PCYT1A | F - GGGCAACAGAAGAAGATGGA | 0.45 | XM_004003005.1 |
| | R - ACCCTGACATAGGGCTTACTA | 0.45 | |
| ABCA3 | F - CCCTTACCCACCTTTCATCTC | 0.45 | XM_004021123.1 |
| | R - CCTTCAGCTTCTTCTCCTTCTC | 0.45 | |
| TTF1 | F - ACACAAAGACCAAACTGCTGGACG | 0.90 | FJ177515 |
| | R - GCGTGGGAAACCCATTTGAATCAC | 0.90 | |
| FOXA1 | F - CGGAGCTTCCAGATTTCTACAC | 0.45 | XM_012098843.2 |
| | R - CCTCGGGCGAAATTCCTAAATA | 0.45 | |
| SP1 | F - TTATCTGCCCAGCCACTTATC | 0.45 | XM_012174188.2 |
| | R - TGCACACTCCAGTGAGTTATC | 0.45 | |
| NR3C1 | F - ACTGCCCCAAGTGAAAACAGA | 0.90 | NM_001114186.1 |
| | R -ATGAACAGAAATGGCAGACATTTTATT | 0.90 | |
| NR3C2 | F - ATGACAGCTCCAAACCAAACACGG | 0.90 | AF349768.1 |
| | R - AAATCCTGGAAGTACCTTCGCCCA | 0.90 | |
| HSD11B1 | F - GCGCCAGATCCCTGTCTGAT | 0.90 | NM_001009395.1 |
| | R - AGCGGGATACCACCTTCTTT | 0.90 | |
| HSD11B2 | F - GAGACATGCCGTTTCCATGC | 0.45 | NM_001009460.1 |
| | R - TGATGCTGACCTTGACACCC | 0.45 | |
| HIF3A | F - GTGGAGTTCCTGGGCATCAG | 0.45 | EU340262.1 |
| | R - CCCGTCAGAAGGAAGCTCAG | 0.45 | |
| EGLN3 | F - TGCTACCCAGGAAATGGAACAGGT | 0.45 | NM_001101164.1 |
| | R - GCTTGGCATCCCAGTTCTTGTTCA | 0.45 | |
| INOS | F - AAGGCAGCCTGTGAGACATT | 0.45 | XM_004012488.1 |
| | R - CAGATTCTGCTGCGATTTGA | 0.45 | |
| ENOS | F - TCTTCCACCAGGAGATGGTC | 0.45 | NM_001129901.1 |
| | R - AGAGGCGTACAGGATGGTTG | 0.45 | |
| SOD1 | F - CTTCGAGGCAAAGGGAGATAAA | 0.45 | FJ546075.1 |
| | R - ACTGGTACAGCCTTGTGTATTG | 0.45 | |
| SOD2 | F - AGTAAACCGTCAGCCTTACAC | 0.45 | NM_001280703.1 |
| | R - CCACGCTCAGAAACACTACA | 0.45 | |
| TFAM | F - GATGATGGAAGTTGGACGAGAA | 0.45 | XM_015104510.1 |
| | R - TACAACAGCTTCGGGTATTGG | 0.45 | |
| NRF1 | F - CGTAGTCCAGACTTTCAGTAACC | 0.45 | XM_012177109.2 |
| | R - ATCAGCAACCGCCGAATAA | 0.45 | |
| HMOX1 | F - CTGGTGATGGCGTCTTTGTA | 0.90 | NM_001014912.1 |
| | R - CAGCTCCTCTGGGAAGTAGA | 0.90 | |
| NOX4 | F - GGCAAGAGAACAGACCTGATTA | 0.45 | EF369489.1 |
| | R - CACCGAGGACGTCCAATAAA | 0.45 | |
| CAT | F - TCACTTTGACCGGGAGAGA | 0.90 | AF236854.1 |
| | R - CGCCTTGGAGTATCTGGTAATG | 0.90 | |
| GPX | F - GTGGCACCATCTATGAGTACG | 0.45 | AF236854.1 |
| | R - CACGTTGACGAAGAGGATGTAT | 0.45 | |

*(Continued)*

**Table 1. (Continued)**

| Primer name | Primer sequence 5′ to 3′ | Primer conc ($\mu$mol $l^{-1}$) | Accession number |
|---|---|---|---|
| *AQP1* | F - AAAGTGTCACTGGCCTTTGGGTTG | 0.45 | NM_001009194.1 |
| | R - ATGTACATGATGGCCCGGAGGATA | 0.45 | |
| *AQP2* | F - TCACTTGAACCCTGCTGTGACCTT | 0.05 | AF123316.1 |
| | R - ACCCGAAGATAATTCCAGCACCCA | 0.05 | |
| *AQP4* | F - TGGGAAATTGGGAGAACCACTGGA | 0.45 | NM_001009279 |
| | R - GGCAGCTTTGCTGAAGGCTTCTTT | 0.45 | |
| *ATP1A1* | F - GGTGTTGCCCTGAGGATGTATC | 0.45 | NM_001009360 |
| | R - CCGGACTTCGTCATACACGAA | 0.45 | |
| *SCNN1A* | F - ACGACAAGAACAGCTCCAACCTCT | 0.90 | AF232715.1 |
| | R - GCCGCAGATTAAAGCCAGCATCAT | 0.90 | |
| *ERN1* | F - CAGGAGTACGTGGAACAGAAG | 0.45 | XM_015098372.1 |
| | R - GGCATGGAGAGGAGGATATTG | 0.45 | |
| *ATF6* | F - AACCACGCAGCTACCTAATC | 0.45 | AY942654.1 |
| | R - CTGTCTCCTTAGCACAGCAATA | 0.45 | |
| *EIF2AK3* | F - CCTTCGGAAGCTTCTCCTTATG | 0.90 | XM_004005901.3 |
| | R - CCGGAGCGCAGTTAGTTTAT | 0.90 | |

| Target protein | Incubation conditions | Molecular wt | Manufacturer |
|---|---|---|---|
| 11$\beta$HSD-2 | 1:1000, in 5% BSA in TBS-T | 44 kDa | Cayman Chemical #10004303 |
| SOD-1 | 1:1000, in 5% BSA in TBS-T | 24 kDa | Sigma–Aldrich #A3854 |
| SP-B | 1:1000, in 5% BSA in TBS-T | 8 kDa | Seven Hills Bioreagents #WRAB-48604 |
| Na+-K+-ATPase-A1 | 1:1000 in 5% BSA in TBS-T | 110 kDa | Invitrogen #MA3-929 |
| TTF-1 | 1:500, in 5% BSA in TBS-T | 38 kDA | Santa Cruz Biotechnology #SC-13040 |
| EGLN1 | 1:500, in 5% BSA in TBS-T | 46 kDa | Novus Biologicals #NB-100-137 |
| EGLN3 | 1:1000, in 5% BSA in TBS-T | 27 kDa | Novus Biologicals #NBPI-30040 |
| Total OXPHOS | 1:1000, in 5% BSA in TBS-T | 20, 30, 40, 48, 55 kDa | Abcam #ab110413 |
| MitoBiogenesis cocktail | 1:1000, in 5% BSA in TBS-T | 35, 70 kDa | Abcam #ab123545 |

markers (*HMOX1, NOX4, CAT, GPX*), water and sodium movement (*AQP1, AQP2, AQP4, ATP1A1, SCNN1A*) and endoplasmic reticulum stress unfolded protein response (*ERN1, ATF6, EIF2AK3*) were measured by qRT-PCR using KiCqStart SYBR Green qPCR ReadyMix (Sigma-Aldrich, USA) in a final volume of 6 $\mu$l on a ViiA7 Fast Real-time PCR system (Applied Biosystems, USA), as previously described (Lock et al., 2017). Each qRT-PCR well contained 3 $\mu$l SYBR Green Master Mix (2X), 2 $\mu$l of forward and reverse primer mixed with $H_2O$ to obtain final primer concentrations and 1 $\mu$l of diluted cDNA. Each sample was run in triplicate for target and reference genes. The abundance of each transcript relative to the abundance of stable reference genes (Hellemans et al., 2007) was calculated using DataAssist 3.0 analysis software (Applied Biosystems, USA) and expressed as mRNA mean normalised expression (MNE) ± SD.

## Quantification of fetal lung protein expression

Protein was extracted by sonication of fetal lung tissue ($\sim$100 mg; NS, $n = 7$; NM, $n = 7$; HS, $n = 6$; HM, $n = 7$) and protein content was determined by a MicroBCA Protein Assay Kit (Thermo Fisher Scientific, Rockford, USA) as previously described (Darby, Sorvina et al., 2020; Lie et al., 2014; Ren et al., 2021). Extracted protein samples were then subjected to sodium dodecyl sulphate (SDS) polyacrylamide gel electrophoresis (PAGE) and stained with Coomassie Blue to ascertain equal protein loading. Protein samples were then transferred onto a 0.45 $\mu$m nitrocellulose membrane (Hybond ECL, NSW, Australia) and subjected to 1 h of drying at room temperature. The membrane was then stained with Ponceau S (0.5% Ponceau in 1% acetic acid) to determine the efficiency of transfer. The membranes were then washed with 7%

acetic acid followed by rinse in RO water and imaged for Ponceau S using an ImageQuant LAS4000 (GE Healthcare, Melbourne, Australia). Following imaging, membranes were washed three times for 5 min in Tris-buffered saline (TBS). The membranes were then blocked with 5% bovine serum albumin (BSA) in TBS with 1% Tween-20 (TBS-T) for 1 h at room temperature. The membranes underwent three washes in TBS-T and were then incubated overnight with the primary antibody at 4°C. The proteins chosen were based on genes that were changed in response to hypoxia and/or MitoQ treatment from the qRT-PCR data to determine if changes in transcription were related to changes in protein expression (Table 1). Following incubation with the primary antibody, membranes were washed and incubated with the appropriate species horseradish peroxidase (HRP) linked secondary antibody for 1 h at room temperature. Enhanced chemiluminescence using SuperSignal West Pico Substrate (Thermo Scientific, Australia) was used to detect blots. Protein abundance was quantified using densitometry using ImageQuant software (GE Healthcare, Victoria, Australia). Total target protein abundance was then normalised to either Ponceau S or to reference protein $\beta$-actin (1:10 000 in 5% BSA in TBS-T, ATCB HRP conjugate, no. 4967, Cell Signalling Technology; 42 kDa band).

### Tissue hormone assay

Tissue hormone concentrations were determined by liquid chromatography (LC; Shimadzu Nexera XR, Shimadzu, Japan) coupled to a SCIEX 6500 Triple-Quad system (MS/MS; SCIEX, US) using an adapted protocol (McBride et al., 2021). Initially, tissue was homogenised in 500 $\mu$l 0.9% NaCl at 50 Hz for 2 min and then centrifuged at 12,000 $g$ for 10 min at 4°C. A100 $\mu$l aliquot of of supernatant was added to 300 $\mu$l acetonitrile containing 50 ng ml$^{-1}$ internal standard (cortisol-9,11,12,12-d4; Toronto Research Chemicals, Toronto, Canada), vortexed for 1 min and then centrifuged at 12,000 $g$ for 10 min. Supernatant was transferred to a fresh Eppendorf tube and the remaining pellet was resuspended in 300 $\mu$l ethyl acetate, vortexed for 1 min and then centrifuged at 12,000 $g$ for 10 min. Supernatant was added to the acetonitrile, mixed by inversion, and then evaporated to dryness using the GeneVac EZ-2 Evaporating System (GeneVac, UK). Dried samples were reconstituted in 50% methanol and then injected onto an ACQUITY UPLC BEH C18 Column 130Å, 1.7 $\mu$m, 2.1 mm × 100 mm (Waters Corp, USA). Mobile phases were 0.1% formic acid in water and 0.1% formic acid in acetonitrile. Flow rate was 0.3 ml min$^{-1}$ and mobile phase B was initially 10% and increased linearly to 90% over 10 min and then held at 90% for 2 min, after which it returned to 10% over 3 min prior to injection of the next sample. Hormone concentrations were calculated via integration with a standard curve that ranged from 0.05 ng ml$^{-1}$ to 100 ng ml$^{-1}$. Conditions for detection of analytes are as previously described (McBride et al., 2021).

### Statistical analysis

Statistical analyses were performed in Graphpad Prism 8 (Graphpad Software Inc., USA). All analyses were assessed for normal distribution of data and a $P < 0.05$ was considered significant. Outliers were determined using the ROUT method (Q = 1%) and removed from the analysis. Analysis of normoxia *versus* hypoxia and saline *versus* MitoQ treatment was performed using a two-way ANOVA to test for an effect of hypoxia, MitoQ and the interaction of these main effects. If a significant interaction was determined, the data were divided by treatment and assessed with a Tukey's *post hoc* test.

## Results

### Fetal biometry

Relative to normoxic pregnancy, fetal body weight and lung weight were both significantly reduced in hypoxic pregnancy at gestational day 138 ($P = 0.0033$ and $P < 0.0001$ respectively, Fig. 1*A* and *B*). There was no significant effect of MitoQ treatment on either fetal body weight, fetal lung weight or in the fetal lung:body weight ratio in normoxic+MitoQ or hypoxic+MitoQ pregnancy (Fig. 1*C*). Fetal arterial blood gas measurements and MitoQ concentration measures have previously been published in Botting et al. (2020). Briefly, exposure of pregnant sheep to a 10% inspired fraction of oxygen led to a controlled reduction by approximately 50% from baseline in the maternal $P_{aO_2}$ (Botting et al., 2020). MitoQ treatment resulted in therapeutic concentrations (>25 pmol g$^{-1}$) in the placenta, fetal skeletal muscle and fetal liver (Botting et al., 2020).

### Expression of genes involved in surfactant protein and phospholipid production

Both hypoxia and MitoQ significantly increased the protein and mRNA expression of *SFTPB* ($P = 0.0071$, 0.0417 and $P = 0.0110$, 0.0291, respectively, Fig. 2*A* and *B*) compared to normoxia and saline, respectively. *SFTPC* mRNA expression was significantly increased by treatment with MitoQ ($P = 0.0290$, Fig. 2*C*) compared to saline. The expression of *SFTPD* was also increased, but only in the hypoxia MitoQ group compared to other treatment groups ($P = 0.0470$, Fig. 2*D*). Relative to normoxic pregnancy, fetal lung *PCYT1A* mRNA

expression was significantly increased in hypoxic fetuses, and this increase was abolished by MitoQ treatment in hypoxic pregnancy ($P = 0.0271$, 0.0249, Fig. 2*E*). The mRNA expression of *ABCA3* and *SFTPA* was not affected by hypoxia or MitoQ (Table 2).

### Expression of genes involved in surfactant protein regulation

There was no significant difference in mRNA expression of *FOXA1* or *SP1* between groups (Table 2). Relative to normoxic pregnancy, *TTF1* mRNA expression was significantly downregulated in the hypoxia saline group, and this change was normalised by MitoQ treatment ($P = 0.0021$, Table 2). Interestingly, protein expression of *TTF1* was significantly increased by hypoxia ($P = 0.0153$, Fig. 2*F*).

### Expression of genes involved in hypoxia signalling and genes involved in nitric oxide synthesis

Relative to normoxic pregnancy, mRNA expression of *HIF3A* and *EGLN3* were significantly increased by hypoxia ($P = 0.0002$ and $P = 0.0003$ respectively, Fig. 3*A* and *D*). *ENOS* mRNA expression was significantly reduced by MitoQ treatment ($P = 0.0121$, Table 2). Hypoxia had a variable effect on the expression of *INOS*, which was downregulated in the saline treated animals as a result of hypoxia but upregulated in the MitoQ treated animals ($P = 0.0495$, Table 2). Protein expression of EGLN1 and ELGN3 were not significantly different between any of the treatment groups (Fig. 3*B* and *C*).

### Expression of genes involved in glucocorticoid signalling and conversion

There was no significant difference in the expression of *NR3C1*, *NR3C2* or *HSD11B1* between groups (Fig. 4 *A–C*). However, relative to normoxic pregnancy, there

was a significant reduction in the mRNA expression of *HSD11B2* caused by hypoxia in addition to an increase caused by MitoQ treatment ($P = 0.0022$, $P = 0.0498$, Fig. 4*D*).

### Expression of mitochondrial oxidative phosphorylation complex components and genes regulating mitochondrial function and transcription factors

We observed no difference in mitochondrial biogenesis via western blot between groups (Fig. 5*A*). The antibody cocktail targets one subunit of Complex IV that includes several proteins encoded by the mitochondrial DNA, and one subunit of Complex II that is entirely encoded in the nucleus. Interestingly, there was reduction in Complex I protein expression in the fetal lung in response to hypoxic pregnancy ($P = 0.0074$, Fig. 5*B*), but no change in Complexes II and IV (Fig. 5*C* and *E*). Relative to saline treated controls, expression of Complex III and of the ATP synthase (Complex V) were significantly increased in the fetal lung as a result of MitoQ treatment ($P = 0.0271$ and $P = 0.0482$, respectively, Fig. 5*D* and *F*). Relative to normoxic pregnancy, *NRF1* mRNA expression was increased by hypoxia with or without MitoQ treatment ($P = 0.0026$, Table 2).

### Expression of pro- and antioxidant markers and genes involved in the endoplasmic reticulum stress unfolded protein response

Relative to normoxic pregnancy, the expression of *NOX4* was decreased in the fetal lungs in treated and untreated hypoxic pregnancy ($P = 0.0081$, Table 2). Relative to normoxic pregnancy, the expression of *ATF6* was decreased in the fetal lungs in hypoxic pregnancy ($P = 0.0483$, Table 2). Conversely, relative to normoxic pregnancy, the expression of *SOD2* was increased in the fetal lungs in treated and untreated hypoxic pregnancy

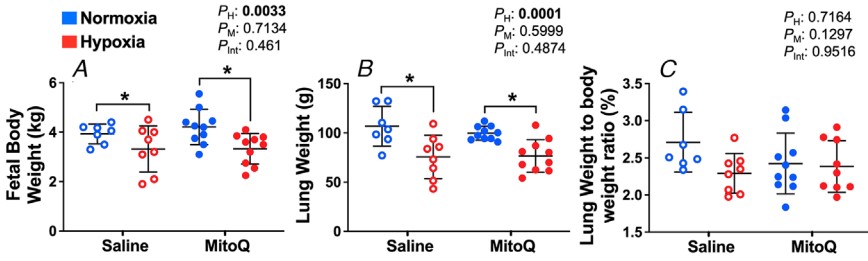

**Figure 1. Fetal biometry**
Data show the mean ± SD with each sample as dot plots for fetal body weight, lung weight and the lung weight to body weight ratio in normoxic (blue symbols) or hypoxic (red symbols) pregnancy with (filled symbols) or without (open symbols) maternal treatment with MitoQ. Data were analysed by two-way ANOVA. If a significant interaction between main factors was found, Tukey's *post hoc* test isolated the significant differences. H, hypoxia main effect, M, MitoQ main effect, Int, interaction. $P < 0.05$ was considered significant. *Effect of hypoxia, †effect of MitoQ.

**Table 2. Effect of hypoxia and MitoQ treatment on expression of genes and proteins regulating surfactant protein and phospholipid production, surfactant protein regulatory genes, glucocorticoid signalling and conversion, hypoxia signalling, nitric oxide synthesis, mitochondrial function and transcription factors, pro- and antioxidant markers, water and sodium movement and endoplasmic reticulum stress unfolded protein response in the fetal lung**

| | Normoxia saline (n = 7) | Normoxia MitoQ (n = 10) | Hypoxia saline (n = 8) | Hypoxia MitoQ (n = 10) | P value Hypoxia | P value MitoQ | P value Interaction |
|---|---|---|---|---|---|---|---|
| **Surfactant protein and phospholipid production** | | | | | | | |
| SFTPA | 0.591 ± 0.276 | 0.567 ± 0.218 | 0.599 ± 0.108 (n = 7) | 0.622 ± 0.223 | 0.6786 | 0.9983 | 0.7597 |
| SFTPB | 1.504 ± 0.343 (n = 6) | 1.753 ± 0.371† (n = 9) | 1.893 ± 0.245* (n = 7) | 2.075 ± 0.581*† | **0.0110** | **0.0291** | 0.8879 |
| SFTPC | 4.164 ± 1.147 (n = 6) | 6.142 ± 1.664† | 5.254 ± 0.807 (n = 7) | 6.032 ± 2.272† | 0.4206 | **0.0290** | 0.3250 |
| SFTPD | 0.030 ± 0.014 | 0.031 ± 0.011 | 0.031 ± 0.011 | 0.044 ± 0.015* (n = 9) | 0.0890 | 0.1147 | **0.0470** |
| PCYT1A | 0.043 ± 0.005 | 0.039 ± 0.004† (n = 9) | 0.049 ± 0.011* | 0.042 ± 0.006*† | **0.0271** | **0.0249** | 0.4881 |
| ABCA3 | 0.093 ± 0.032 | 0.102 ± 0.025 | 0.101 ± 0.028 | 0.116 ± 0.030 | 0.2708 | 0.2051 | 0.7428 |
| SP-B (AU) | 0.0220 ± 0.0108 (n = 7) | 0.0263 ± 0.0146† (n = 7) | 0.0339 ± 0.0320* (n = 6) | 0.0704 ± 0.0333*† (n = 6) | **0.0071** | **0.0417** | 0.1023 |
| **Surfactant protein regulatory genes** | | | | | | | |
| TTF1 | 0.151 ± 0.059 (n = 6) | 0.115 ± 0.032 | 0.091 ± 0.013* (n = 6) | 0.149 ± 0.038 | 0.3528 | 0.4637 | **0.0021** |
| FOXA1 | 0.001 ± 0.0003 | 0.001 ± 0.0003 | 0.001 ± 0.0002 | 0.001 ± 0.0003 | 0.2761 | 0.9364 | 0.3199 |
| SP1 | 0.281 ± 0.032 | 0.236 ± 0.040 | 0.262 ± 0.034 | 0.261 ± 0.039 | 0.8343 | 0.0779 | 0.0951 |
| TTF-1 (AU) | 0.123 ± 0.0607 (n = 7) | 0.130 ± 0.0580 (n = 7) | 0.217 ± 0.0967* (n = 6) | 0.175 ± 0.0602* (n = 7) | **0.0153** | 0.5425 | 0.1753 |
| **Glucocorticoid signalling and conversion** | | | | | | | |
| NR3C1 | 0.200 ± 0.031 | 0.176 ± 0.031 | 0.184 ± 0.027 | 0.198 ± 0.025 | 0.7433 | 0.6208 | 0.0578 |
| NR3C2 | 0.008 ± 0.002 | 0.006 ± 0.001 (n = 9) | 0.007 ± 0.002 | 0.007 ± 0.002 | 0.6041 | 0.4561 | 0.3493 |
| HSD11B1 | 0.013 ± 0.005 | 0.012 ± 0.005 | 0.013 ± 0.003 | 0.014 ± 0.004 | 0.4456 | 0.9681 | 0.5636 |
| HSD11B2 | 0.003 ± 0.001 | 0.004 ± 0.001† | 0.002 ± 0.001* (n = 7) | 0.003 ± 0.0004*† | **0.0022** | **0.0498** | 0.5106 |
| **hypoxia signalling** | | | | | | | |
| HIF3A | 0.031 ± 0.008 (n = 6) | 0.030 ± 0.007 (n = 8) | 0.047 ± 0.013* | 0.056 ± 0.020* | **0.0002** | 0.4210 | 0.3416 |
| EGLN3 | 0.008 ± 0.003 (n = 6) | 0.012 ± 0.006 | 0.020 ± 0.009* (n = 6) | 0.024 ± 0.010* | **0.0003** | 0.1210 | 0.9642 |
| EGLN1 (AU) | 0.0070 ± 0.0062 (n = 7) | 0.0063 ± 0.0023 (n = 7) | 0.0065 ± 0.0063 (n = 6) | 0.0046 ± 0.0042 (n = 7) | 0.7822 | 0.6994 | 0.9805 |
| EGLN3 (AU) | 0.0547 ± 0.0055 (n = 7) | 0.0506 ± 0.0070 (n = 7) | 0.055 ± 0.0084 (n = 6) | 0.0517 ± 0.0050 (n = 7) | 0.7164 | 0.1297 | 0.9516 |
| **Nitric oxide synthesis** | | | | | | | |
| INOS | 0.053 ± 0.011 (n = 6) | 0.048 ± 0.010 (n = 7) | 0.041 ± 0.005* | 0.057 ± 0.010* (n = 9) | 0.3912 | 0.4095 | **0.0495** |
| ENOS | 0.017 ± 0.002 | 0.014 ± 0.003† | 0.016 ± 0.004 (n = 9) | 0.014 ± 0.003† | 0.4957 | **0.0121** | 0.4829 |
| **Mitochondrial function and transcription factors** | | | | | | | |
| TFAM | 0.067 ± 0.011 | 0.061 ± 0.010 | 0.067 ± 0.008 | 0.072 ± 0.006 | 0.4254 | 0.0615 | 0.0132 |
| NRF1 | 0.036 ± 0.002 (n = 6) | 0.031 ± 0.006 | 0.039 ± 0.006* | 0.041 ± 0.005* (n = 9) | **0.0026** | 0.3052 | 0.1061 |
| Complex I (AU) | 0.0078 ± 0.0037 (n = 7) | 0.0090 ± 0.0017 (n = 7) | 0.0054 ± 0.0015* (n = 6) | 0.0057 ± 0.0022* (n = 6) | **0.0074** | 0.4461 | 0.6385 |
| Complex II (AU) | 0.0023 ± 0.0006 (n = 7) | 0.0028 ± 0.0003 (n = 7) | 0.0022 ± 0.0003 (n = 6) | 0.0023 ± 0.0008 (n = 7) | 0.1908 | 0.1338 | 0.5472 |
| Complex III (AU) | 0.0057 ± 0.0017 (n = 7) | 0.0076 ± 0.0009† (n = 7) | 0.0054 ± 0.0008 (n = 6) | 0.0063 ± 0.0020† (n = 7) | 0.1655 | **0.0271** | 0.3980 |
| Complex IV (AU) | 0.0004 ± 0.0001 (n = 7) | 0.0005 ± 0.0002 (n = 7) | 0.0003 ± 6e-005 (n = 6) | 0.0004 ± 0.0002 (n = 7) | 0.0855 | 0.2949 | 0.5726 |
| Complex V (AU) | 0.0063 ± 0.0019 (n = 7) | 0.0075 ± 0.0013† (n = 7) | 0.0056 ± 0.0010 (n = 6) | 0.0067 ± 0.0023† (n = 7) | 0.2547 | **0.0482** | 0.8942 |
| Mitobiogenesis | 0.1300 ± 0.0458 (n = 7) | 0.1606 ± 0.0563 (n = 7) | 0.1361 ± 0.0486 (n = 6) | 0.1449 ± 0.0834 (n = 7) | 0.8393 | 0.4096 | 0.6461 |

*(Continued)*

**Table 2. (Continued)**

| | Normoxia saline (n = 7) | Normoxia MitoQ (n = 10) | Hypoxia saline (n = 8) | Hypoxia MitoQ (n = 10) | P value Hypoxia | P value MitoQ | P value Interaction |
|---|---|---|---|---|---|---|---|
| Pro and antioxidant markers | | | | | | | |
| HMOX1 | 0.037 ± 0.012 | 0.032 ± 0.009[†] (n = 9) | 0.039 ± 0.010 | 0.029 ± 0.008[†] (n = 9) | 0.8961 | **0.0480** | 0.4241 |
| NOX4 | 0.004 ± 0.002 | 0.005 ± 0.002 | 0.003 ± 0.001* | 0.003 ± 0.001* | **0.0081** | 0.4408 | 0.5246 |
| CAT | 0.151 ± 0.025 | 0.150 ± 0.027 | 0.158 ± 0.017 (n = 7) | 0.145 ± 0.029 | 0.8951 | 0.4464 | 0.5295 |
| GPX | 0.013 ± 0.005 | 0.010 ± 0.003 (n = 9) | 0.016 ± 0.007 | 0.013 ± 0.003 (n = 9) | 0.0986 | 0.0897 | 0.7261 |
| SOD1 | 0.337 ± 0.070 | 0.330 ± 0.073 (n = 9) | 0.348 ± 0.009 (n = 6) | 0.316 ± 0.027 | 0.9319 | 0.3118 | 0.5177 |
| SOD2 | 0.064 ± 0.018 | 0.064 ± 0.014 | 0.077 ± 0.013* | 0.088 ± 0.015* | **0.0006** | 0.5257 | 0.1564 |
| SOD (AU) | 0.0114 ± 0.0033 (n = 7) | 0.0122 ± 0.0035 (n = 7) | 0.0092 ± 0.0026 (n = 6) | 0.0106 ± 0.0021 (n = 7) | 0.3217 | 0.1092 | 0.8114 |
| Water and sodium movement | | | | | | | |
| AQP1 | 0.126 ± 0.020 (n = 6) | 0.146 ± 0.029 (n = 9) | 0.190 ± 0.049* (n = 7) | 0.165 ± 0.037* | **0.0032** | 0.8661 | **0.0497** |
| AQP2 | 0.0004 ± 0.00003 (n = 5) | 0.0004 ± 0.0001 | 0.0006 ± 0.0002* (n = 7) | 0.0006 ± 0.0002* | **0.0043** | 0.3744 | 0.6583 |
| AQP4 | 0.010 ± 0.001 (n = 5) | 0.013 ± 0.008 (n = 9) | 0.015 ± 0.009 | 0.014 ± 0.008 | 0.3357 | 0.6460 | 0.4681 |
| ATP1A1 | 0.051 ± 0.015 (n = 5) | 0.058 ± 0.008 (n = 9) | 0.055 ± 0.015* | 0.058 ± 0.011 | 0.2521 | 0.0567 | **0.0435** |
| SCNN1A | 0.015 ± 0.005 | 0.013 ± 0.004 (n = 9) | 0.018 ± 0.005 (n = 7) | 0.021 ± 0.007* (n = 9) | 0.2529 | 0.6906 | **0.0313** |
| ATP1A1 (AU) | 0.0798 ± 0.0317 (n = 7) | 0.0653 ± 0.0256[†] (n = 7) | 0.0869 ± 0.0355 (n = 6) | 0.0382 ± 0.0114[†] (n = 7) | 0.3506 | **0.0062** | 0.1166 |
| Endoplasmic reticulum stress unfolded protein response | | | | | | | |
| ERN1 | 0.014 ± 0.002 (n = 6) | 0.015 ± 0.003 | 0.014 ± 0.002 | 0.016 ± 0.005 | 0.2824 | 0.4214 | 0.5545 |
| ATF6 | 0.149 ± 0.041 (n = 6) | 0.128 ± 0.043 (n = 8) | 0.104 ± 0.008* (n = 6) | 0.132 ± 0.043 | 0.1600 | 0.8137 | 0.0975 |
| EIF2AK3 | 0.035 ± 0.007 | 0.028 ± 0.007 | 0.033 ± 0.004 | 0.034 ± 0.003 | 0.2548 | 0.1044 | **0.0483** |

Data are expressed as either as mean normalized expression (MNE) or as arbitrary units (AU) relative to Ponceau S ± SD and were analysed by two-way ANOVA. $P < 0.05$ was considered significant (values in bold). *Effect of hypoxia, [†]effect of MitoQ.

($P = 0.0006$, Table 2). The mRNA expression of *HMOX1, CAT* and *GPX* was not significantly changed by hypoxia or MitoQ treatment (Table 2). Values for the *ERN1* and *EIF2AK3, SOD1* and *TFAM* mRNA expression were also not significantly different between any of the treatment groups (Table 2).

### Expression of genes important for water and sodium movement

The mRNA expression of *AQP2* and *AQP4* was unaffected by hypoxia or MitoQ treatment (Table 2). Relative to normoxic pregnancy, values for *ATP1A1* and *AQP1* mRNA expression were both significantly increased in the hypoxia saline group ($P = 0.0062$ and $P = 0.0032$, respectively, Fig. 6B and C). However, MitoQ treatment in hypoxic pregnancy normalised this difference. The *SCNN1A* mRNA expression was increased in the

hypoxia MitoQ treated animals compared to normoxia MitoQ ($P = 0.0313$, Fig. 6D). Interestingly, the ATP1A1 protein expression was decreased by MitoQ treatment in the fetal lungs, the opposite of mRNA expression, which may imply post-transcriptional regulation of this gene ($P = 0.0062$, Fig. 6A).

### Concentration of hormones within fetal lung tissue

There was no significant effect of hypoxia or MitoQ treatment on tissue concentration of cortisol or cortisone (Table 3). Relative to normoxia controls, there was a reduction in lung tissue hormone concentration of triiodothyronine ($T_3$) and thyroxine ($T_4$) in hypoxic pregnancy ($P = 0.0320$ and $P = 0.0189$, respectively, Table 3). Lung tissue progesterone concentration was significantly reduced in MitoQ treated animals compared to saline controls ($P = 0.0094$, Table 3).

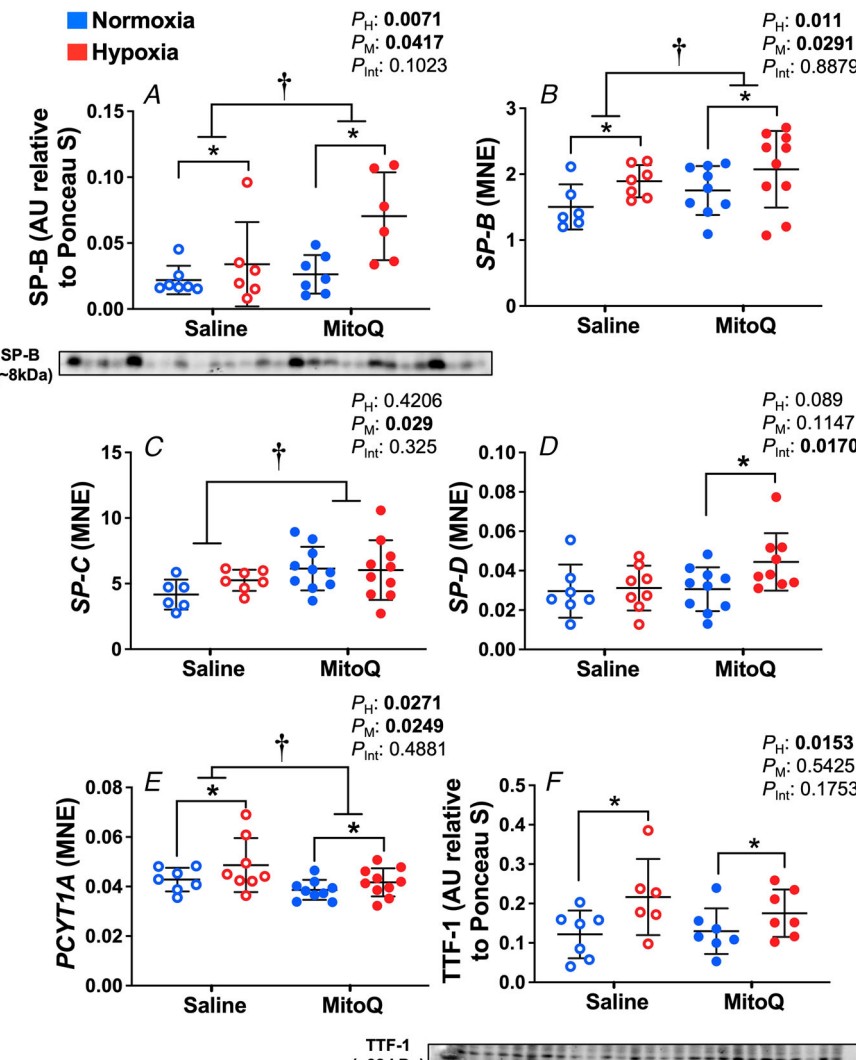

**Figure 2. Expression of surfactant protein B, surfactant protein gene expression and genes regulating surfactant phospholipid and protein production in the fetal lung**
Data show the mean ± SD with each sample as dot plots for the expression of surfactant protein B (SP-B), *SP-C* and *SP-D* and regulators of surfactant protein and phospholipid production; *PCYT1A* and *TTF1* in normoxic (blue symbols) or hypoxic (red symbols) pregnancy with (filled symbols) or without (open symbols) maternal treatment with MitoQ. Data are expressed as arbitrary units (AU) relative to Ponceau S, mRNA mean normalised expression (MNE). Data were analysed by two-way ANOVA. If a significant interaction between main factors was found, Tukey's *post hoc* test isolated the significant differences. H, h = Hypoxia main effect, M, MitoQ main effect, Int, interaction. $P < 0.05$ was considered significant. *Effect of hypoxia, †effect of MitoQ.

## Discussion

Maternal treatment with antioxidants is one of the most promising pharmacological therapies for preventing oxidative stress in fetuses exposed to hypoxic conditions *in utero* (Botting et al., 2020; Hansell et al., 2022; Niu et al., 2018; Spiroski et al., 2021; for review, see Giussani, 2021). Previous studies have reported a potential therapeutic benefit of conventional antioxidant treatment during pregnancy, such as with vitamin C, for fetal lung development in hypoxic pregnancies (McGillick et al., 2021). However, vitamin C is a weak antioxidant and high doses incompatible with human treatment are needed for effective *in vivo* therapy (Botting et al., 2020; Giussani,

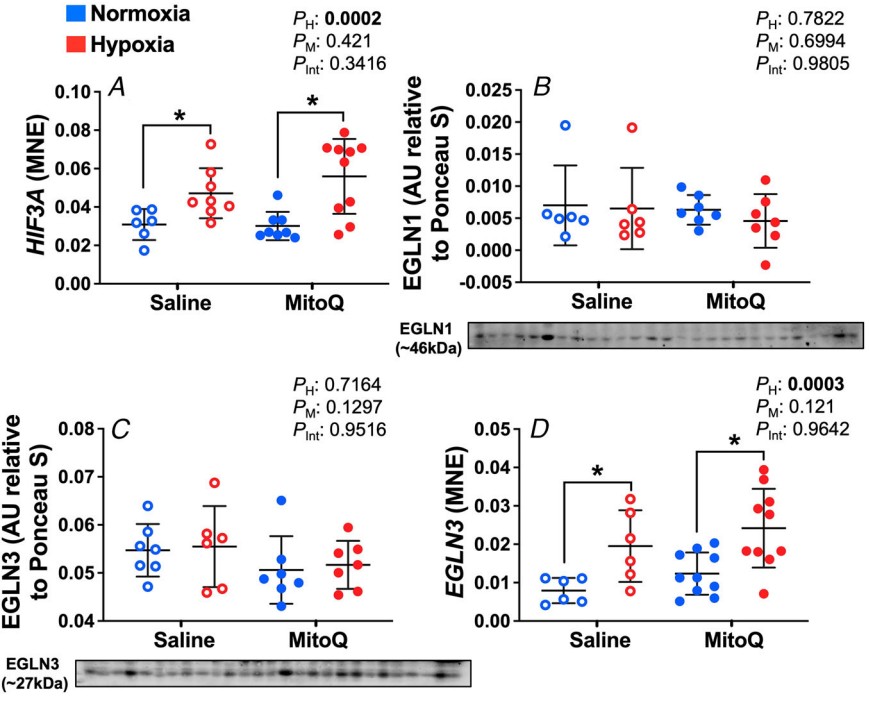

**Figure 3. Protein expression of hypoxia markers in the fetal lung**
Data show the mean ± SD with each sample as dot plots for the expression of *HIF3A*, EGLN1 and EGLN3 in normoxic (blue symbols) or hypoxic (red symbols) pregnancy with (filled symbols) or without (open symbols) maternal treatment with MitoQ. Data are expressed as; arbitrary units (AU) relative to Ponceau S or mRNA mean normalised expression (MNE) and were analysed by two-way ANOVA. If a significant interaction between main factors was found, Tukey's *post hoc* test isolated the significant differences. H, hypoxia main effect, M, MitoQ main effect, Int, interaction. $P < 0.05$ was considered significant. *Effect of hypoxia, †effect of MitoQ.

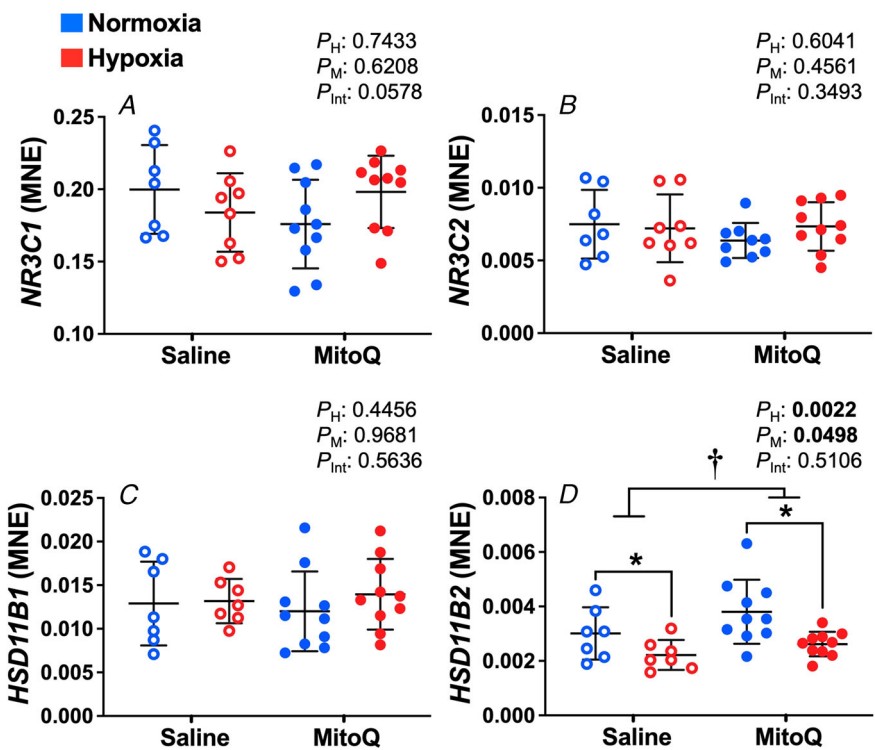

**Figure 4. Expression of genes regulating glucocorticoid signalling and cortisol/cortisone conversion**
Data show the mean ± SD with each sample as dot plots for the expression of genes regulating glucocorticoid signalling; *NR3C1*, *NR3C2* and conversion of cortisol and cortisone; *HSD11B1, HSD11B2* in normoxic (blue symbols) or hypoxic (red symbols) pregnancy with (filled symbols) or without (open symbols) maternal treatment with MitoQ. Data are expressed as mean normalised expression (MNE) and were analysed by two-way ANOVA. If a significant interaction between main factors was found, Tukey's *post hoc* test isolated the significant differences. H, hypoxia main effect, M, MitoQ main effect, Int, interaction. $P < 0.05$ was considered significant. *Effect of hypoxia, †effect of MitoQ.

2021; Giussani et al., 2012). Here we show maturational effects on the fetal lung of a mitochondria-targeted antioxidant therapy at doses appropriate for human clinical translation.

The effect of fetal hypoxaemia on lung development is variable and it depends on the timing, severity and duration of the insult (Darby, Varcoe et al., 2020; McGillick et al., 2017; Morrison, 2008). For instance, while pulmonary surfactant maturation is significantly reduced in a sheep model of early-onset FGR during gestation (Orgeig et al., 2010), it is significantly increased in ovine models of late-onset chronic fetal hypoxaemia during gestation (Gagnon et al., 1999; McGillick et al., 2017). In the present study, chronic hypoxia in the last third of pregnancy also increased markers of surfactant maturation in the fetal lung. This may be explained by the increase in the protein expression of thyroid transcription factor-1 (*TTF-1*) in lungs of hypoxic fetuses, as *TTF-1* is known to upregulate the expression of SPs during lung development (Morrison et al., 2012). In the present study, there was also an increase in the expression of *PCYT1A* mRNA expression in lungs of hypoxic fetal sheep. PCYT1A is a rate limiting enzyme in *de novo* synthesis of surfactant phospholipid production, indicating that in addition to promoting SP expression, hypoxia may also have a positive impact on surfactant phospholipid production resulting in an increase in surfactant insertion into the alveolar hypophase. In addition, in the present study, maternal treatment with MitoQ acted together with hypoxia to substantially increase the expression of SPs (Fig. 2). Importantly, there were no detrimental effects of MitoQ treatment alone on the molecular components measured in the present study, suggesting that maternal antioxidant does not impair the normal maturation of any of the examined pathways within the study.

Hypoxic pregnancy caused an increase in expression of genes involved in hypoxia signalling in the fetal lung. This included an increase in *HIF3A*, a key regulator of late branching morphogenesis, alveolar formation and epithelial differentiation, thereby playing a critical role in proximal airway development (Huang et al., 2013). We also measured an increase in the expression of genes with hypoxia-response elements within their promotor regions, such as *EGLN3* (also known as PHD3) in response to hypoxia. Due to the relatively short half-life of the HIF-$\alpha$ protein subunits, we were unable

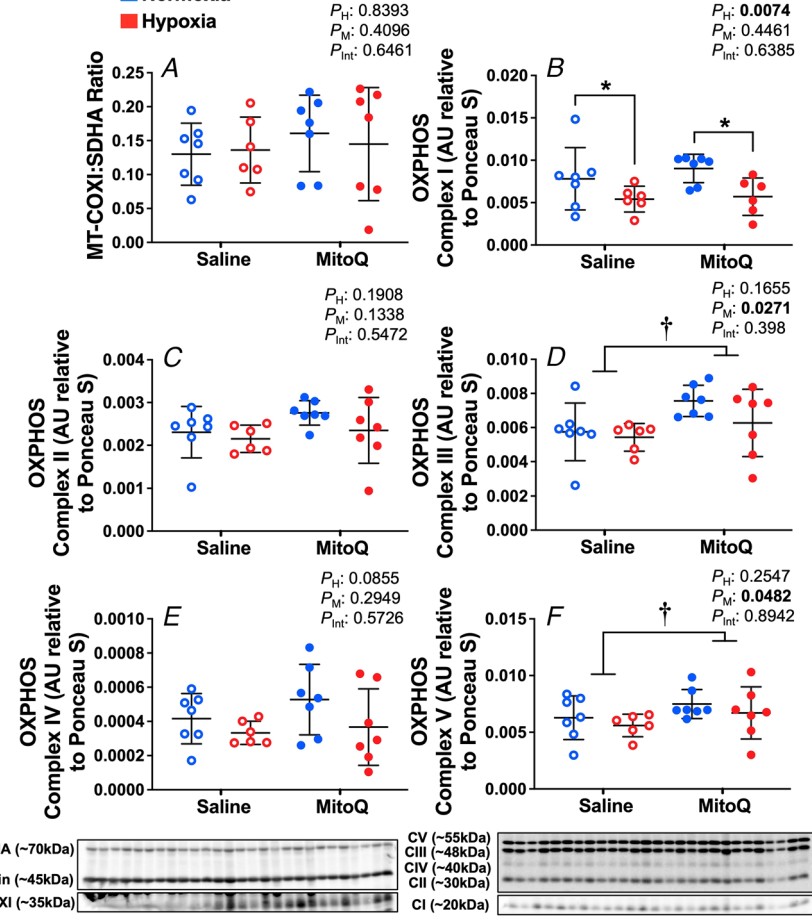

**Figure 5. Expression of mitochondrial complexes and mitobiogenesis**

Data show the mean ± SD with each sample as dot plots for mitobiogenesis (COX-I:SDH-A ratio) and the expression of mitochondrial Complex I, II, III, IV and V in normoxic (blue symbols) or hypoxic (red symbols) pregnancy with (filled symbols) or without (open symbols) maternal treatment with MitoQ. Data are expressed as arbitrary units (AU) relative to Ponceau S and were analysed by two-way ANOVA. If a significant interaction between main factors was found, Tukey's *post hoc* test isolated the significant differences. H, hypoxia main effect, M, MitoQ main effect, Int, interaction. $P < 0.05$ was considered significant. *Effect of hypoxia, †effect of MitoQ.

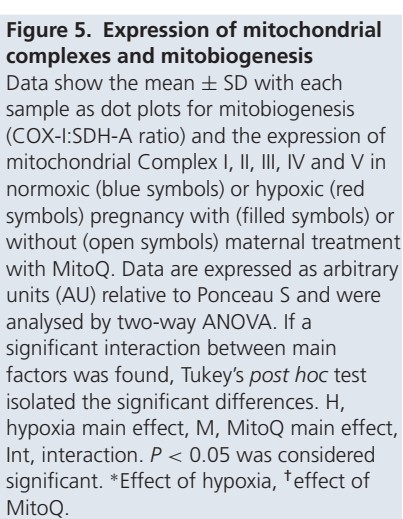

to investigate protein expression directly. Interestingly, when investigating the expression of PHD1 (*EGLN1)* and PHD3 (*EGLN3)*, there was no impact of hypoxia or MitoQ at the protein level. Previous studies using this ovine model found that maternal treatment with vitamin C increased the expression of *EGLN3* (McGillick et al., 2021). However, in the current study, maternal treatment with MitoQ had no effect on *EGLN3* signalling in the fetal lung. This differential outcome may be due to the mode of action between conventional and mitochondria-targeted antioxidants.

Interestingly, when interrogating the expression of sodium movement, there appears to be some post-transcriptional modulation of gene expression occurring within the fetal lung. Although *ATP1A1* mRNA expression was upregulated in fetal lungs from hypoxic pregnancies with no effect on MitoQ treated animals, there was a significant decrease in protein expression caused by MitoQ. Differences in mRNA and protein expression in this case may be due to modulation of gene expression by miRNA intervention. Maternal treatment with MitoQ also significantly increased the expression of sodium and water movement genes *SCNN1A* and *AQP*, respectively. These genes are not only important in the regulation of active fetal lung liquid reabsorption before birth, but also in the basal regulation of liquid movement at the air-tissue interface in the air breathing lung. Taken together, these data suggest that at the genomic level MitoQ may confer some benefit in promoting genes important in the transition to air-breathing at birth.

The increase in fetal plasma cortisol concentration that occurs in most species including humans close to term is largely responsible for preparing the lung for the trans-ition to air-breathing at birth (Fowden et al., 1998). In the present study, although the bulk of genes involved in glucocorticoid regulation were not significantly impacted by either hypoxia or MitoQ treatment, there was a significant decrease in *11BHSD2* mRNA expression in the fetal lung in hypoxic pregnancy, and an increase in *11BHSD2* mRNA expression in pregnancy treated with MitoQ. These changes in glucocorticoid regulation indicate that hypoxia may increase glucocorticoid bioactivity by reducing the conversion of bioactive cortisol to bio-inactive cortisone, potentially explaining the maturational effects of hypoxic pregnancy on the fetal lung. Conversely, the opposing effect of MitoQ treatment on the fetal lung suggests a potential decrease in cortisol availability in the fetal lung. However, the lack of change in tissue cortisol and cortisone concentration and the expression of other regulators of local glucocorticoid bioactivity and availability suggests that there is little overall impact of either hypoxia or MitoQ treatment on maturational effects mediated by cortisol within the fetal lung.

Since MitoQ is a targeted antioxidant that uses the cell and the mitochondrial membrane potential to accumulate within the mitochondria (Botting et al., 2020), we measured changes in the protein expression of members of the mitochondrial electron transport chain (ETC) and in ATP synthase. Additional data in the present study show a significant decrease in Complex I expression in the fetal lung in response to hypoxic pregnancy, while a significant increase in the expression of Complex III and Complex V (ATP Synthase) in the fetal lung following maternal treatment with MitoQ. Complex I is one of the main sites for superoxide anion generation and downregulation

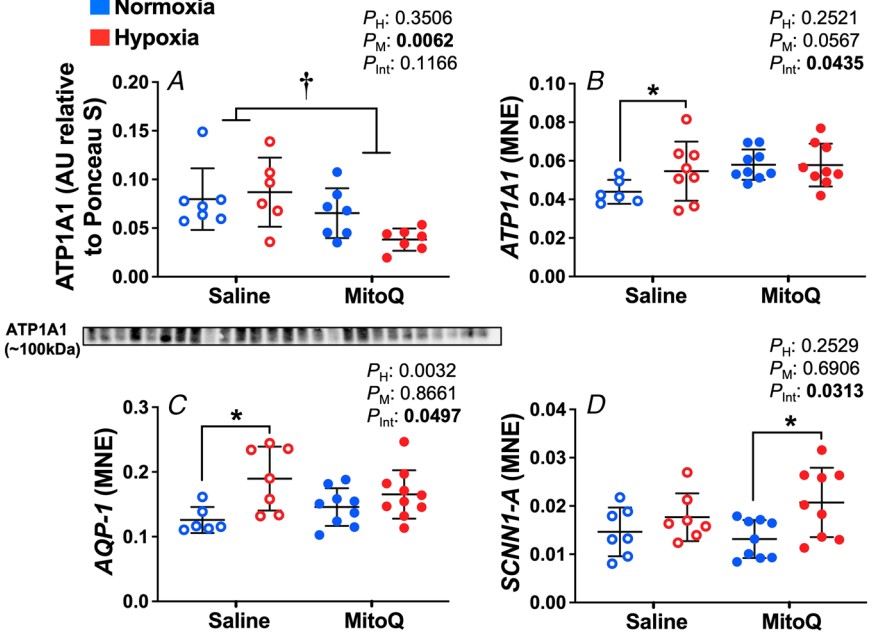

**Figure 6. Expression of genes regulating sodium and water movement in the fetal lung**
Data show the mean ± SD with each sample as dot plots for the expression of genes regulating sodium; ATP1A1, *SCNN1-A* and water movement; *AQP-1*, in normoxic (blue symbols) or hypoxic (red symbols) pregnancy with (closed symbols) or without (open symbols) maternal treatment with MitoQ. Data are expressed as; arbitrary units (AU) relative to Ponceau S, mRNA mean normalised expression (MNE). Data were analysed by two-way ANOVA. If a significant interaction between main factors was found, Tukey's *post hoc* test isolated the significant differences. H, hypoxia main effect, M, MitoQ main effect, Int, interaction. *P* < 0.05 was considered significant. *Effect of hypoxia, †effect of MitoQ.

**Table 3. Effect of hypoxia and MitoQ treatment on the concentration of hormones within the fetal lung**

| | Normoxia saline (n = 7) | Normoxia MitoQ (n = 6) | Hypoxia saline (n = 6) | Hypoxia MitoQ (n = 7) | P value Hypoxia | P value MitoQ | P value Interaction |
|---|---|---|---|---|---|---|---|
| Cortisol (ng ml⁻¹ mg⁻¹) | $0.0016 \pm 0.0010$ | $0.0021 \pm 0.0006$ | $0.0023 \pm 0.0013$ | $0.0027 \pm 0.0013$ | 0.1369 | 0.3076 | 0.9726 |
| Cortisone (ng ml⁻¹ mg⁻¹) | $0.0036 \pm 0.0009$ | $0.0027 \pm 0.0010$ (n = 5) | $0.0031 \pm 0.0009$ | $0.0029 \pm 0.0007$ | 0.7025 | 0.1848 | 0.3407 |
| Progesterone (ng ml⁻¹ mg⁻¹) | $0.0046 \pm 0.0018$ (n = 6) | $0.0026 \pm 0.0001$† | $0.0035 \pm 0.0021$ | $0.0023 \pm 0.0007$† | 0.2327 | **0.0094** | 0.5620 |
| T₃ (ng ml⁻¹ mg⁻¹) | $0.0004 \pm 6e{-}005$ (n = 6) | $0.0004 \pm 4e{-}005$ | $0.0004 \pm 4e{-}005$* | $0.0003 \pm 6e{-}005$* | **0.0320** | 0.2756 | 0.9392 |
| T₄ (ng ml⁻¹ mg⁻¹) | $0.0247 \pm 0.0143$ | $0.0142 \pm 0.0029$ | $0.0098 \pm 0.0028$* | $0.0124 \pm 0.0059$* | **0.0189** | 0.2386 | 0.0575 |

Data were analysed by two-way ANOVA. $P < 0.05$ was considered significant (values in bold). *Effect of hypoxia. †effect of MitoQ.

of its activity is an established endogenous antioxidant defence that limits free radical synthesis (Heather et al., 2012; Horscroft et al., 2017; Lesnefsky et al., 2004; Murphy & Steenbergen, 2008; Sack, 2006). This compensatory antioxidant response has been reported in other hypoxic tissues including human skeletal muscle and placenta, as well as in the adult rat heart (Colleoni et al., 2013; Heather et al., 2012; Horscroft et al., 2017). A differential effect of hypoxic pregnancy on different complexes also supports that this is not the result of a general fall in mitochondrial number or bioactivity; rather the compensatory antioxidant defence is specific to Complex I in the hypoxic fetal lung. A lack of change in mitochondrial biogenesis is also supported by the lack of effect on the COX-I:SDH-A ratio. In fact, the significant increase in the expression of *NRF1* in the fetal lung in response to hypoxia may be a compensatory response to increase mitochondrial biogenesis. *NRF1* is a transcription factor that functions primarily as a positive regulator of nuclear genes involved in mitochondrial biogenesis and oxidative phosphorylation, such as *TFAM* (Scarpulla, 2008). An increase in Complex III and Complex V expression in the fetal lung in pregnancy treated with MitoQ supports an increased capacity to pump protons against their electrochemical gradient to enhance the proton motive force that drives ATP synthesis.

The lung antioxidant system in the mature fetus close to term is vitally important to defend against the rapid change in oxygen tension at birth, and the corresponding increase in the generation of ROS. Therefore, changes in the balance of expression of antioxidant and pro-oxidant enzymes in the fetal lung are crucial. Additional data in the present study show that hypoxic pregnancy increased the expression of *SOD2* mRNA, decreased the expression of *NOX4* and *ATF6*, but it did not affect the expression of *CAT* or *GP*X in the fetal lung. The NADPH oxidase 4 is a pro-oxidant enzyme encoded by the *NOX4* gene (Harijith et al., 2022). *ATF6* is a transcription factor involved in the activation of the unfolded protein response and shown to be increased in the placenta of hypoxic pregnancy (Tong et al., 2022). Therefore, a fall in the expression of *NOX4* and *ATF6* coupled with an increase in the antioxidant *SOD2* mRNA suggest a compensatory increase in the antioxidant balance in the fetal lung. However, mitochondrial function and ROS production were not determined in this study due to the requirement for fresh tissue samples, and this may be an important avenue to explore in future studies. This limitation was due to tissues being generated as part of a programme of work designed with the primary objective of investigating cardiovascular physiology in the offspring (Botting et al., 2020; Brain et al., 2019) and on the placenta (Tong et al., 2022). Therefore, only effects of hypoxic pregnancy with and without maternal treatment with MitoQ on fetal lung molecular studies could be determined, without corroborating pulmonary function

studies or assessment of wet/dry lung ratio. Lung tissue was not instillation fixed to preserve airway structure, and therefore airspace ratio and quantification of type II alveolar epithelial cells was also not able to be assessed. The observed increase in SP expression may therefore be due to an increase in differentiation/larger population of type II alveolar epithelial cells, rather than an upregulation of SP production within the existing cells. An additional limitation of the work is that only male fetuses were used in this study, as females were assigned to postnatal studies (Botting et al., 2020; Brain et al., 2019), so we are unable to interrogate potential sex differences.

## Conclusion

It is currently unknown how the use of antioxidants during pregnancy affects the developing fetal lung in humans. Using an animal model of increased human translational potential, here we show that treatment of healthy or hypoxic pregnancy with the targeted antioxidant MitoQ, increases the expression of key molecules involved in surfactant maturation, lung liquid reabsorption and in mitochondrial proteins driving ATP synthesis in the fetal sheep lung (see Abstract figure). Therefore, MitoQ may be a useful targeted antioxidant therapy to help accelerate fetal lung maturation in healthy and complicated pregnancies.

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

## Additional information

### Data availability statement

All data supporting the results are presented in the manuscript.

### Competing interests

The authors have no conflict of interest to disclose.

### Author contributions

M.C.L., S.O., D.A.G. and J.L.M. were responsible for the conception and design of the experiments. M.C.L., K.J.B., B.J.A., Y.N., S.G.F. and D.A.G. performed experiments. M.C.L., D.A.G. and J.L.M. analysed the data. M.C.L., K.J.B., S.O., D.A.G. and J.L.M. drafted the paper. All authors edited and approved the paper. D.A.G. and J.L.M. obtained funding. All authors agree to be accountable for all aspects of the work. All persons designated as authors qualify for authorship, and all those who qualify for authorship are listed.

### Funding

The animal work was funded by a programme grant (RG/17/8/32924) from The British Heart Foundation (D.A.G.). D.A.G. is Professor of Cardiovascular Physiology & Medicine at the Department of Physiology Development & Neuroscience at the University of Cambridge, Professorial Fellow and Director of Studies in Medicine at Gonville & Caius College, a Lister Institute Fellow and a Royal Society Wolfson Research Merit Award Holder. J.L.M. and the molecular work were funded by an ARC Future Fellowship (Level 3; FT170100431). M.P.M. is funded by the Medical Research Council UK (MC_UU_00028/4).

### Acknowledgements

We are grateful to the University Biomedical Services staff of the University of Cambridge and the Early Origins of Adult Health Research Group, Emma Bradshaw, Melanie Bertossa and Stacey Holman, in performing the western blots.

Open access publishing facilitated by University of South Australia, as part of the Wiley - University of South Australia agreement via the Council of Australian University Librarians.

### Keywords

antioxidant, fetal growth restriction, fetus, hypoxia, lung development, surfactant

### Supporting information

Additional supporting information can be found online in the Supporting Information section at the end of the HTML view of the article. Supporting information files available:

**Statistical Summary Document**
**Peer Review History**

