## [Peer Review History · The Journal of Physiology]

MitoQ as an antenatal antioxidant treatment improves markers of lung maturation in healthy and hypoxic pregnancy

Mitchell C Lock, Kimberley J Botting, Beth J Allison, Youguo Niu, Sage Ford, Michael P. Murphy, Sandra Orgeig, Dino A Giussani, and Janna L Morrison

DOI: 10.1113/JP284786

Corresponding author(s): Janna Morrison (janna.morrison@unisa.edu.au)

The following individual(s) involved in review of this submission have agreed to reveal their identity: Prudence Pereira (Referee #1)

Review Timeline:

Submission Date:	03-Apr-2023
Editorial Decision:	18-May-2023
Revision Received:	08-Jun-2023
Editorial Decision:	21-Jun-2023
Revision Received:	25-Jun-2023
Editorial Decision:	29-Jun-2023
Revision Received:	30-Jun-2023
Accepted:	04-Jul-2023

Senior Editor: Laura Bennet

Reviewing Editor: Christopher Lear

Transaction Report:

Dear Janna,

Re: JP-RP-2023-284786 "MitoQ as an antenatal antioxidant treatment improves lung development in healthy and hypoxic pregnancy" by Mitchell C Lock, Kimberley J Botting, Beth J Allison, Youguo Niu, Sage Ford, Michael P. Murphy, Sandra Orgeig, Dino A Giussani, and Janna L Morrison

Thank you for submitting your manuscript to The Journal of Physiology. It has been assessed by a Reviewing Editor and by 2 expert referees and we are pleased to tell you that it is potentially acceptable for publication following satisfactory major revision.

REVISION CHECKLIST:

We look forward to receiving your revised submission.

Best wishes,

Professor Laura Bennet
Senior Editor
The Journal of Physiology
<https://jp.msubmit.net>
<http://jp.physoc.org>
The Physiological Society
Hodgkin Huxley House
30 Farringdon Lane
London, EC1R 3AW
UK
<http://www.physoc.org>
<http://journals.physoc.org>

REQUIRED ITEMS

-Author photo and profile. First (or joint first) authors are asked to provide a short biography (no more than 100 words for one author or 150 words in total for joint first authors) and a portrait photograph. These should be uploaded and clearly labelled with the revised version of the manuscript. See Information for Authors for further details.

-You must start the Methods section with a paragraph headed Ethical Approval. A detailed explanation of journal policy and regulations on animal experimentation is given in Principles and standards for reporting animal experiments in The Journal of Physiology and Experimental Physiology by David Grundy J Physiol, 593: 2547-2549. doi:10.1113/JP270818.). A checklist outlining these requirements and detailing the information that must be provided in the paper can be found at: <https://physoc.onlinelibrary.wiley.com/hub/animal-experiments>. Authors should confirm in their Methods section that their experiments were carried out according to the guidelines laid down by their institution's animal welfare committee, and conform to the principles and regulations as described in the Editorial by Grundy (2015). The Methods section must contain details of the anaesthetic regime: anaesthetic used, dose and route of administration and method of killing the experimental animals.

-Please upload separate high-quality figure files via the submission form.

-Please ensure that any tables are in Word format and are, wherever possible, embedded in the article file itself.

-Please ensure that the Article File you upload is a Word file.

-A Statistical Summary Document, summarising the statistics presented in the manuscript, is required upon revision. It must be on the Journal's template, which can be downloaded from the link in the Statistical Summary Document section here: https://jp.msubmit.net/cgi-bin/main.plex?form_type=display_requirements#statistics

-Papers must comply with the Statistics Policy https://jp.msubmit.net/cgi-bin/main.plex?form_type=display_requirements#statistics

In summary:

- If $n \leq 30$, all data points must be plotted in the figure in a way that reveals their range and distribution. A bar graph with data points overlaid, a box and whisker plot or a violin plot (preferably with data points included) are acceptable formats.
- If $n > 30$, then the entire raw dataset must be made available either as supporting information, or hosted on a not-for-profit repository e.g. FigShare, with access details provided in the manuscript.
- n clearly defined (e.g. x cells from y slices in z animals) in the Methods. Authors should be mindful of pseudoreplication.
- All relevant n values must be clearly stated in the main text, figures and tables, and the Statistical Summary Document (required upon revision)
- The most appropriate summary statistic (e.g. mean or median and standard deviation) must be used. Standard Error of the Mean (SEM) alone is not permitted.
- Exact p values must be stated. Authors must not use 'greater than' or 'less than'. Exact p values must be stated to three significant figures even when 'no statistical significance' is claimed.
- Statistics Summary Document completed appropriately upon revision

-Please include an Abstract Figure file, as well as the figure legend text within the main article file. The Abstract Figure is a piece of artwork designed to give readers an immediate understanding of the research and should summarise the main conclusions. If possible, the image should be easily 'readable' from left to right or top to bottom. It should show the physiological relevance of the manuscript so readers can assess the importance and content of its findings. Abstract Figures should not merely recapitulate other figures in the manuscript. Please try to keep the diagram as simple as possible and without superfluous information that may distract from the main conclusion(s). Abstract Figures must be provided by authors no later than the revised manuscript stage and should be uploaded as a separate file during online submission labelled as File Type 'Abstract Figure'. Please ensure that you include the figure legend in the main article file. All Abstract Figures should be created using BioRender. Authors should use The Journal's premium BioRender account to export high-resolution images. Details on how to use and access the premium account are included as part of this email.

EDITOR COMMENTS

Reviewing Editor:

The clear strength of the present study is the model utilised to induce hypoxic pregnancy and the ability to assess the impact of MitoQ on multiple outcomes, as the Cambridge group have previously published. The main weakness however is that no data on pulmonary function or structure are provided to support the physiological impact of the changes in mRNA and protein expression. The manuscript would be substantially improved and more likely to proceed if the authors are able to expand their analysis and provide direct evidence of the effects of MitoQ on lung maturation or function. The reviewers have raised several additional concerns that need to be addressed.

Attention is needed to include exact p values in figures, text and tables.

Other points

Line 314 - please rephrase to acknowledge no effect on the components examined rather than suggesting all components.

As noted by the reviewers, blots are required to be included.

The inclusion of only male fetuses is inappropriate.

REFEREE COMMENTS

Referee #1:

Thankyou for the opportunity to review this timely manuscripts. Mitochondrial based therapies and antioxidants are

increasingly being recognized as the new frontier in neonatal medicine. As such this is a timely and well conducted study.

A generally well detailed methods section. Could the authors please provide additional detail on where lung samples were obtained from (i.e. left or right lobes) and if the sampling sight was standardized across all animals. I commend the authors on their extensive testing of the reference genes but note that the RT-PCR section does not currently meet MIQE guidelines (as stated by the authors). To meet the essential MIQE checks I request that the authors include the number of samples analyzed by RT-PCR in each group, and in a table the primer sequences and reaction conditions. For general readability the primary antibody details could be included as a supplementary Table. For non sheep researchers it would be useful for the authors to include reference to what the full gestation period for the species would be (i.e. ~145d).

The authors are to be commended for their thoroughness regarding the number of proteins or genes that were measured. I do feel however that some tweaking may improve the impact of the results. For example, in the graphs it would be useful where both a western blot and PCR result is available for the same target to always have it in the order protein-PCR (there are swaps between figure 2 and 3 for example). I also found the use of the 'over' comparison in Figure 2A,B,C,E etc to be confusing. In terms of the post-hoc analysis which comparison groups do these represent? It would be useful if the same formatting used to compare the normoxia vs hypoxia for saline or MitoQ was used to show the significant comparisons between all blue or all red comparisons.

Not essential but would have been useful to demonstrate the impact of mitoQ treatment on cell composition within the lung. Are the increases in SP-B due to more AT2 cells, or because the AT2 cells are induced by hypoxia and/or mitoQ to produce more protein? This could be included as a limitation within the discussion.

I am a little hesitant by the authors use of the the statement "Here we show maturational effects on the fetal lung of a mitochondria-targeted antioxidant therapy..." you were able to detect evidence of maturation such as SP-B, C, D production etc in all groups treated with saline, and without histological evidence I'm not sure there is a strong evidence that lung maturation was delayed by hypoxia. Rather much of the evidence is that mitoQ increases cellular function of some targets. The authors also alude to this in para 1, page 6.

Referee #2:

Lock M and colleagues investigated in their study MitoQ as a possible preventive strategy for chronic fetal hypoxemia. To this end, ewes were treated with MitoQ during late gestation and offspring's lungs were analysed with a focus on pulmonary surfactant maturation, mitochondrial respiration, ATP synthesis, and lung liquid reabsorption. The data demonstrate significant effects of hypoxia on lung maturation. MitoQ, however, showed only a mild preventive effect.

The research question is of great interest, the topic clinically very relevant, and the model appropriate. However, the data do not show marked effect of MitoQ on lung maturation after chronic fetal hypoxia. Overall, some additional aspects should be considered to strengthen the conclusion made by the authors:

- 1) The title implies a stronger effect of MitoQ than the data ultimately show. The authors might consider to rephrase.
- 2) The introduction is somewhat lengthy and could be more focused. There are some sections that belong in the discussion rather than in the introduction. This could certainly be improved to emphasize the focus of the study.
- 3) Animal protocol approval number should be included. Why did they only include male fetuses in this study? Did they expect sex-specific differences/response to hypoxia and/ MitoQ?
- 4) I am uncertain what the purpose of the surgical intervention was? The overall aim was to combine chronic hypoxia with MitoQ treatment. Why was the operation performed at day 100 of gestation?
- 5) Was the pharmacological dose of MitoQ tested? Were markers of oxidative stress as a readout for sufficient MitoQ effect

investigated (e.g., 8-Oxo-dG, ROS,...)?

6) The authors only show the protein quantification. The blots with bands for the respective protein and loading should be shown.

7) Was the wet lung/dry lung ratio measured as an indicator of impaired lung liquid absorption?

8) The authors show effects of hypoxia/MitoQ on the gene expression of surfactant proteins; was the protein expression assessed as well?

9) Were hypoxia-induced changes in lung structure attenuated by MitoQ?

10) Since the authors focus on surfactant proteins, it would be interesting to see if fraction of AT2 is changed by hypoxia+/- MitoQ.

11) Was lung cell proliferation or apoptosis assessed?

END OF COMMENTS

Confidential Review

03-Apr-2023

“MitoQ as an antenatal antioxidant treatment improves lung development in healthy and hypoxic pregnancy”

Manuscript: JP-RP-2023-284786

Dear Prof Bennet,

We thank the reviewing editor and reviewers for their critical appraisal of our manuscript. Please see below a detailed response to each comment.

EDITOR COMMENTS

Reviewing Editor:

1) The clear strength of the present study is the model utilised to induce hypoxic pregnancy and the ability to assess the impact of MitoQ on multiple outcomes, as the Cambridge group have previously published. The main weakness however is that no data on pulmonary function or structure are provided to support the physiological impact of the changes in mRNA and protein expression. The manuscript would be substantially improved and more likely to proceed if the authors are able to expand their analysis and provide direct evidence of the effects of MitoQ on lung maturation or function. The reviewers have raised several additional concerns that need to be addressed.

We thank the Reviewing Editor for their careful consideration and critique of the manuscript. Unfortunately, as the tissue used in this study was being generated as part of a programme of work designed with the primary objective of investigating cardiovascular physiology in the offspring (Brain *et al.*, 2019; Botting *et al.*, 2020) and on the placenta (Tong *et al.*, 2022), no lung function studies were performed. Thus, only effects of hypoxic pregnancy with and without maternal treatment with MitoQ on fetal lung molecular studies could be determined. In addition, lung structure could not be assessed as the tissue was not appropriately instillation fixed to preserve airway structure. Despite the lack of functional data, these studies provide novel information of important physiological relevance in an ovine model of high translational relevance. The data in this paper would support the rationale for such functional analysis in future studies. In addition, we undertook these studies in line with the 3Rs to reduce the number of animals that undergo studies, and best make use of the valued experimental material. We have altered the title of the manuscript and expanded the *Discussion* section on technical limitations to address the Reviewing Editor's comment.

Line 401: *“However, mitochondrial function and ROS production were not determined in this study due to the requirement of fresh tissue samples, and this may be an important avenue to explore in future studies. This limitation was due to tissues being generated as part of a programme of work designed with the primary objective of investigating cardiovascular physiology in the offspring (Brain et al., 2019; Botting et al., 2020) and on the placenta (Tong et al., 2022). Therefore, only effects of hypoxic pregnancy with and without maternal treatment with MitoQ on fetal lung molecular studies could be determined, without corroborating pulmonary function studies, or assessment of wet/dry lung ratio. Lung tissue was not instillation fixed to preserve airway structure, and therefore airspace ratio and quantification of type II alveolar epithelial cells was also not able to be assessed. The observed increase in surfactant protein expression may therefore be due to an increase in differentiation/larger population of type II alveolar epithelial cells, rather than an upregulation of surfactant protein production within the existing cells.”*

2) Attention is needed to include exact p values in figures, text and tables.

We thank the Reviewing Editor for this correction. P values have now been included for all values in text and in data tables.

3) Line 314 - please rephrase to acknowledge no effect on the components examined rather than suggesting all components.

We have adjusted phrasing to improve clarity.

Line 324: "Importantly, there were no detrimental effects of MitoQ treatment alone on the molecular components measured in the present study, suggesting that maternal antioxidant does not impair the normal maturation of any of the examined pathways within the study."

4) As noted by the reviewers, blots are required to be included.

We apologise for this omission. Images of the western blots have now been included within the figures.

5) The inclusion of only male fetuses is inappropriate.

To make the study viable ethically and economically, every singleton ovine pregnancy generated was used as in previous studies (McGillick *et al.*, 2017; Botting *et al.*, 2020; McGillick *et al.*, 2021).

Therefore, studies in the fetal period used the male offspring, while studies in the adult period used the female offspring, as ewe lambs are easier to group house and maintain compared to rams. Female fetuses were used for postnatal experiments (Brain *et al.*, 2019).

REFeree COMMENTS

Referee #1:

Thank you for the opportunity to review this timely manuscript. Mitochondrial based therapies and antioxidants are increasingly being recognized as the new frontier in neonatal medicine. As such this is a timely and well conducted study.

1) A generally well detailed methods section. Could the authors please provide additional detail on where lung samples were obtained from (i.e. left or right lobes) and if the sampling sight was standardized across all animals.

We thank the Reviewer for their praise and support of our study. All lung samples were collected from the lower right lobe and sampled from the same relative position from each animal. We have added this detail to the *Methods* section for clarity.

Line 133: "Lung tissue for molecular analysis was collected from the lower right lobe from the same position in each fetus and immediately frozen in liquid nitrogen for qRT-PCR and western blot analyses."

2) I commend the authors on their extensive testing of the reference genes but note that the RT-PCR section does not currently meet MIQE guidelines (as stated by the authors). To meet the essential MIQE checks I request that the authors include the number of samples analyzed by RT-PCR in each group, and in a table the primer sequences and reaction conditions.

We thank the Reviewer for indicating the requirements to meet the MIQE guidelines and apologise for this omission. The sample size for each individual primer has now been included in Table 1. The primer sequences have all previously been published within our group and are referenced within the *Methods* section. However, to address the Reviewer's comment, we have now included all sequences within Table 1 for clarity.

3) For general readability the primary antibody details could be included as a supplementary Table. For non sheep researchers it would be useful for the authors to include reference to what the full gestation period for the species would be (i.e. ~145d).

Thank you. The primary antibody details have now been included within Table 1. The length of gestation is stated at the beginning of the *Methods* section.

Line 83: "Pregnant ewes carrying a singleton pregnancy (determined by ultrasound scan at 80 days of gestation; term, 145 days) underwent surgery under general anaesthesia using aseptic conditions at 100 ± 1 days of gestation."

4) The authors are to be commended for their thoroughness regarding the number of proteins or genes that were measured. I do feel however that some tweaking may improve the impact of the results. For example, in the graphs it would be useful where both a western blot and PCR result is available for the same target to always have it in the order protein-PCR (there are swaps between figure 2 and 3 for example).

We thank the Reviewer for their suggestion and have changed the graphs to reflect protein expression before mRNA expression for the same target as a standard throughout the manuscript. We feel that this suggestion has markedly improved the presentation of the data.

5) I also found the use of the 'over' comparison in Figure 2A,B,C,E etc to be confusing. In terms of the post-hoc analysis which comparison groups do these represent? It would be useful if the same formatting used to compare the normoxia vs hypoxia for saline or MitoQ was used to show the significant comparisons between all blue or all red comparisons.

Thank you for the suggestion. We have changed the graphs to use the same symbols in Table 2 to more clearly display the significant changes per treatment.

6) Not essential but would have been useful to demonstrate the impact of mitoQ treatment on cell composition within the lung. Are the increases in SP-B due to more AT2 cells, or because the AT2 cells are induced by hypoxia and/or mitoQ to produce more protein? This could be included as a limitation within the discussion.

Thank you for the discussion point. Unfortunately, we were unable to assess the numerical density of type II AECs within this study. However, it is true that the observed increase in surfactant protein expression may be due to an increase in differentiation/larger population of type II AECs, rather than an upregulation of production within the existing cells. To address the Reviewer's point, this has been added to the *Discussion* section as a technical limitation of the study.

Line 408: "Lung tissue was not instillation fixed to preserve airway structure, and therefore airspace ratio and quantification of type II alveolar epithelial cells was also not able to be assessed. The observed increase in surfactant protein expression may therefore be due to an increase in differentiation/larger population of type II alveolar epithelial cells, rather than an upregulation of surfactant protein production within the existing cells."

7) I am a little hesitant by the authors use of the statement "Here we show maturational effects on the fetal lung of a mitochondria-targeted antioxidant therapy..." you were able to detect evidence of maturation such as SP-B, C, D production etc in all groups treated with saline, and without histological evidence I'm not sure there is a strong evidence that lung maturation was delayed by

hypoxia. Rather much of the evidence is that mitoQ increases cellular function of some targets. The authors also allude to this in para 1, page 6.

Thank you. We agree. We have softened the relevant statements to more closely reflect the findings of the study. As mentioned in the *Discussion* section, the effect of fetal hypoxaemia on lung development is variable and depends on the timing, severity and duration of the insult. Using the current model of chronic hypoxia in the last third of pregnancy, our results are consistent with our previous work, showing increased pulmonary surfactant maturation (McGillick *et al.*, 2017). This is in contrast to early-onset fetal growth restriction where surfactant maturation is impaired. It is therefore important to interrogate the effectiveness of MitoQ in a number of models of complicated pregnancy to determine if the same beneficial outcomes are observed.

Line 70: “Here, we tested the hypothesis that maternal treatment with MitoQ in late gestation will improve markers of maturation in the developing lung in both normal and hypoxic pregnancy in sheep, a species with similar fetal lung developmental milestones as humans (Lock et al., 2013; Morrison et al., 2018).”

Line 305: “Here we show maturational effects on the fetal lung of a mitochondria-targeted antioxidant therapy at doses appropriate for human clinical translation.”

Referee #2:

Lock M and colleagues investigated in their study MitoQ as a possible preventive strategy for chronic fetal hypoxemia. To this end, ewes were treated with MitoQ during late gestation and offspring's lungs were analysed with a focus on pulmonary surfactant maturation, mitochondrial respiration, ATP synthesis, and lung liquid reabsorption. The data demonstrate significant effects of hypoxia on lung maturation. MitoQ, however, showed only a mild preventive effect.

The research question is of great interest, the topic clinically very relevant, and the model appropriate. However, the data do not show marked effect of MitoQ on lung maturation after chronic fetal hypoxia. Overall, some additional aspects should be considered to strengthen the conclusion made by the authors:

1) The title implies a stronger effect of MitoQ than the data ultimately show. The authors might consider to rephrase.

We thank the Reviewer for their suggestion. We have modified the title to reflect the study more accurately.

“MitoQ as an antenatal antioxidant treatment improves markers of lung maturation in healthy and hypoxic pregnancy”

2) The introduction is somewhat lengthy and could be more focused. There are some sections that belong in the discussion rather than in the introduction. This could certainly be improved to emphasize the focus of the study.

We thank the Reviewer for their critique. To address the Reviewer’s comment, we have reduced the length of the *Introduction* by removing an entire paragraph and focusing previous statements.

3) Animal protocol approval number should be included. Why did the only include male fetuses in this

study? Did they expect sex-specific differences/response to hypoxia and/ MitoQ?

Thank you for these important comments. We have now added the appropriate animal ethics approval numbers to the manuscript.

Line 74: “Experimental protocols for animal research were performed under the UK Animals (Scientific Procedures) Act 1986 and were approved by the Ethical Review Committee of the University of Cambridge under Home Office Project Licence PL70/7645 and PL80/2232.”

To make the study viable ethically and economically, every singleton ovine pregnancy generated was used. Therefore, studies in the fetal period used the male offspring, while studies in the adult period used the female offspring, as ewe lambs are easier to group house and maintain compared to rams. Female fetuses were used for other postnatal experiments (Brain et al., 2019).

4) I am uncertain what the purpose of the surgical intervention was? The overall aim was to combine chronic hypoxia with MitoQ treatment. Why was the operation performed at day 100 of gestation?

The surgical procedure was used to determine fetal sex to allow allocation to the present fetal component of the project if male (as presented within this study) or used for postnatal studies if female (as ewes are easier to group house than rams). Surgery was also performed to catheterise the maternal femoral artery and vein so that daily blood samples could be taken from the maternal artery. The degree of hypoxia within the chamber was titrated based on the PaO₂ of the mother, to achieve 50 mmHg. MitoQ treatment or saline were administered daily into the venous catheter.

5) Was the pharmacological dose of MitoQ tested? Were markers of oxidative stress as a readout for sufficient MitoQ effect investigated (e.g., 8-Oxo-dG, ROS,...)?

Measurement of MitoQ content in placentomes, fetal mid-brain, fetal biceps femoris, and fetal liver were performed using LC-MS/MS as previously described (Botting et al., 2020). To address the Reviewer’s comment, the latter study has been referenced within the manuscript.

Line 229: “MitoQ treatment resulted in therapeutic concentrations (>25pmol g⁻¹) in the placenta, fetal skeletal muscle, and fetal liver (Botting et al., 2020).”

Maternal daily MitoQ treatment, from 105 to 138 days of gestation, resulted in greater therapeutic concentrations of MitoQ in the placenta, fetal skeletal muscle, and fetal liver than with single administration. Though oxidative stress was not assessed in these sheep, cardiac mitochondrial respiratory control ratio was assessed in hypoxic chicken embryos where it prevented the enhanced *in vivo* mitochondria-derived oxidative stress in the embryonic heart and restored the cardiac mitochondrial respiratory control ratio, left ventricular structure, and systolic function (Botting et al., 2020).

6) The authors only show the protein quantification. The blots with bands for the respective protein and loading should be shown.

We apologise. Western blot images have now been included within figures.

7) Was the wet lung/dry lung ratio measured as an indicator of impaired lung liquid absorption?

The tissue used in this study was being generated as part of a programme of work designed with the primary objective of investigating cardiovascular physiology in the offspring (Brain et al., 2019; Botting et al., 2020) or effects on the placenta (Tong et al., 2022). As collection of lung tissue was not the primary outcome, we were unable to assess wet lung/dry lung ratio within this study and were limited to

only molecular studies. We have expanded on this within the *Discussion* section as a technical limitation to address the Reviewer's comment.

Line 406: Therefore, only effects of hypoxic pregnancy with and without maternal treatment with MitoQ on fetal lung molecular studies could be determined, without corroborating pulmonary function studies, or assessment of wet/dry lung ratio.

8) The authors show effects of hypoxia/MitoQ on the gene expression of surfactant proteins; was the protein expression assessed as well?

SP-B protein expression is displayed in Figure 2A alongside the corresponding mRNA expression. Unfortunately, due to limitations regarding antibody specificity for sheep, we were unable to assess other surfactant proteins within this study. In our previous studies, we have shown that changes in mRNA and protein expression have a consistent pattern of expression (Orgeig *et al.*, 2010).

9) Were hypoxia-induced changes in lung structure attenuated by MitoQ?

Lung development was not the primary outcome of the project. Thus, only effects of hypoxic pregnancy with and without maternal treatment with MitoQ on fetal lung molecular studies could be determined within this study. Lung structure could not be assessed as the tissue was not appropriately instillation fixed to preserve airway structure. To address the Reviewer's comment, we have expanded on this within the *Discussion* section as a technical limitation.

Line 408: "Lung tissue was not instillation fixed to preserve airway structure, and therefore airspace ratio and quantification of type II alveolar epithelial cells was also not able to be assessed."

10) Since the authors focus on surfactant proteins, it would be interesting to see if fraction of AT2 is changed by hypoxia+/-MitoQ.

Thank you for the discussion point. Unfortunately, we were unable to assess the numerical density of type II AECs within this study. It is true however that the observed increase in surfactant protein expression may be due to an increase in differentiation/larger population of type II AECs, rather than an upregulation of production within the existing cells. To address the Reviewer's comment, this has been added to the *Discussion* section as a technical limitation of the study.

Line 408: "Lung tissue was not instillation fixed to preserve airway structure, and therefore airspace ratio and quantification of type II alveolar epithelial cells was also not able to be assessed. The observed increase in surfactant protein expression may therefore be due to an increase in differentiation/larger population of type II alveolar epithelial cells, rather than an upregulation of surfactant protein production within the existing cells."

11) Was lung cell proliferation or apoptosis assessed?

We were particularly interested in surfactant development within this study and without being able to assess the density of type II AECs as discussed above, we were also unable to assess proliferation or apoptosis of these cells within the fetal lungs due to a lack of instillation fixed tissue. To our knowledge there is no evidence of developmental hypoxia leading to apoptosis within the fetal lung.

References

- Botting KJ, Skeffington KL, Niu Y, Allison BJ, Brain KL, Itani N, Beck C, Logan A, Murray AJ, Murphy MP & Giussani DA. (2020). Translatable mitochondria-targeted protection against programmed cardiovascular dysfunction. *Science Advances* **6**, eabb1929.
- Brain KL, Allison BJ, Niu Y, Cross CM, Itani N, Kane AD, Herrera EA, Skeffington KL, Botting KJ & Giussani DA. (2019). Intervention against hypertension in the next generation programmed by developmental hypoxia. *PLOS Biology* **17**, e2006552.
- Lock M, McGillick EV, Orgeig S, McMillen IC & Morrison JL. (2013). Regulation of fetal lung development in response to maternal overnutrition. *Clinical and experimental pharmacology & physiology* **40**, 803-816.
- McGillick EV, Orgeig S, Allison BJ, Brain KL, Niu Y, Itani N, Skeffington KL, Kane AD, Herrera EA, Giussani DA & Morrison JL. (2017). Maternal chronic hypoxia increases expression of genes regulating lung liquid movement and surfactant maturation in male fetuses in late gestation. *The Journal of Physiology* **595**, 4329-4350.
- McGillick EV, Orgeig S, Allison BJ, Brain KL, Niu Y, Itani N, Skeffington KL, Kane AD, Herrera EA, Morrison JL & Giussani DA. (2021). Molecular regulation of lung maturation in near-term fetal sheep by maternal daily vitamin C treatment in late gestation. *Pediatric Research*.
- Morrison JL, Berry MJ, Botting KJ, Darby JRT, Frasch MG, Gatford KL, Giussani DA, Gray CL, Harding R, Herrera EA, Kemp MW, Lock MC, McMillen IC, Moss TJ, Musk GC, Oliver MH, Regnault TRH, Roberts CT, Soo JY & Tellam RL. (2018). Improving pregnancy outcomes in humans through studies in sheep. *American Journal of Physiology-Regulatory, Integrative and Comparative Physiology* **315**, R1123-r1153.
- Orgeig S, Crittenden TA, Marchant C, McMillen IC & Morrison JL. (2010). Intrauterine growth restriction delays surfactant protein maturation in the sheep fetus. *American Journal of Physiology-Lung Cellular and Molecular Physiology* **298**, L575-583.
- Tong W, Allison BJ, Brain KL, Patey OV, Niu Y, Botting KJ, Ford SG, Garrud TA, Wooding PFB, Shaw CJ, Lyu Q, Zhang L, Ma J, Cindrova-Davies T, Yung HW, Burton GJ & Giussani DA. (2022). Chronic Hypoxia in Ovine Pregnancy Recapitulates Physiological and Molecular Markers of Preeclampsia in the Mother, Placenta, and Offspring. *Hypertension* **79**, 1525-1535.

Dear Janna,

Re: JP-RP-2023-284786R1 "MitoQ as an antenatal antioxidant treatment improves markers of lung maturation in healthy and hypoxic pregnancy" by Mitchell C Lock, Kimberley J Botting, Beth J Allison, Youguo Niu, Sage Ford, Michael P. Murphy, Sandra Orgeig, Dino A Giussani, and Janna L Morrison

Thank you for submitting your manuscript to The Journal of Physiology. It has been assessed by a Reviewing Editor and by 2 expert referees and we are pleased to tell you that it is acceptable for publication following satisfactory revision.

REVISION CHECKLIST:

We look forward to receiving your revised submission.

Best wishes,

Professor Laura Bennet
Senior Editor
The Journal of Physiology
<https://jp.msubmit.net>
<http://jp.physoc.org>
The Physiological Society
Hodgkin Huxley House
30 Farringdon Lane
London, EC1R 3AW
UK
<http://www.physoc.org>
<http://journals.physoc.org>

EDITOR COMMENTS

Reviewing Editor:

Thank you for your revised manuscript, the reviewers have no further issues that need to be addressed. I however do have some minor issues to be addressed.

Regardless of the inclusion of p values in figure legends, figures 1-6 can very easily be adapted to show the exact p values rather than summary symbols, as is the preferred style of the Journal.

The number of subjects varies in some results, eg Table 2. Please provide a brief explanation likely in methodology for why n has been reduced in some analyses.

Abstract: "maternal antioxidant therapy in hypoxic pregnancy has proven to be protective with regards to fetal growth and cardiovascular development" Please rewrite to be more cautious or clarify this evidence is from animal studies.

REFEREE COMMENTS

Referee #1:

I thank the authors for their detailed response to the review. On extensive review it appears the authors have addressed and made changes in relations to the original reviewing comments appropriately.

Referee #2:

I have no further comments.

END OF COMMENTS

1st Confidential Review

08-Jun-2023

“MitoQ as an antenatal antioxidant treatment improves markers of lung maturation in healthy and hypoxic pregnancy”

Manuscript: JP-RP-2023-284786R1

Dear Prof Bennet,

We thank the reviewing editor and reviewers for their approval of our manuscript. Please see below a detailed response to the remaining editor comments.

EDITOR COMMENTS

Reviewing Editor:

Regardless of the inclusion of p values in figure legends, figures 1-6 can very easily be adapted to show the exact p values rather than summary symbols, as is the preferred style of the Journal.

The P values have now been included on each graph as requested.

The number of subjects varies in some results, eg Table 2. Please provide a brief explanation likely in methodology for why n has been reduced in some analyses.

The missing samples were outliers that were removed from the analysis; this detail was previously included within the qRT-PCR method section but has now been moved to a more appropriate location within the statistical analysis methodology.

Line 217: “Outliers were determined using the ROUT method (Q=1%) and removed from the analysis.”

Abstract: "maternal antioxidant therapy in hypoxic pregnancy has proven to be protective with regards to fetal growth and cardiovascular development" Please rewrite to be more cautious or clarify this evidence is from animal studies.

Thank you for the suggestion, we have changed this sentence to clarify that the evidence is referring to animal studies.

Line 2: “In turn, chronic fetal hypoxaemia promotes oxidative stress, and maternal antioxidant therapy in animal models of hypoxic pregnancy has proven to be protective with regards to fetal growth and cardiovascular development.”

Dear Professor Morrison,

Re: JP-RP-2023-284786R2 "MitoQ as an antenatal antioxidant treatment improves markers of lung maturation in healthy and hypoxic pregnancy" by Mitchell C Lock, Kimberley J Botting, Beth J Allison, Youguo Niu, Sage Ford, Michael P. Murphy, Sandra Orgeig, Dino A Giussani, and Janna L Morrison

Thank you for submitting your revised Research Article to The Journal of Physiology. It has been assessed by the original Reviewing Editor and Referees and has been well received. Some final additions have been requested. Please see Required Items Section below.

We hope that you will be able to return your revised manuscript within one week. If you require longer than this, please contact journal staff: jp@physoc.org.

We look forward to receiving your revised submission.

Yours sincerely,

Professor Laura Bennet
Senior Editor
The Journal of Physiology
<https://jp.msubmit.net>
<http://jp.physoc.org>
The Physiological Society
Hodgkin Huxley House
30 Farringdon Lane
London, EC1R 3AW
UK
<http://www.physoc.org>
<http://journals.physoc.org>

REQUIRED ITEMS FOR REVISION

-You must upload original, uncropped western blot/gel images (including controls) if they are not included in the manuscript. This is to confirm that no inappropriate, unethical or misleading image manipulation has occurred <https://physoc.onlinelibrary.wiley.com/hub/journal-policies#imagmanip> These should be uploaded as 'Supporting information for review process only'. Please label/highlight the original gels so that we can clearly see which sections/lanes have been used in the manuscript figures.

-Please include an Abstract Figure legend text within the main article file.

EDITOR COMMENTS

Reviewing Editor:

Thank you for your alterations.

END OF COMMENTS

2nd Confidential Review

25-Jun-2023

“MitoQ as an antenatal antioxidant treatment improves markers of lung maturation in healthy and hypoxic pregnancy”

Manuscript: JP-RP-2023-284786R2

We thank the reviewing editor for their thorough review of our manuscript. Please see below a detailed response to the remaining editor comments.

EDITOR COMMENTS

REQUIRED ITEMS FOR REVISION

-You must upload original, uncropped western blot/gel images (including controls) if they are not included in the manuscript. This is to confirm that no inappropriate, unethical or misleading image manipulation has occurred <https://physoc.onlinelibrary.wiley.com/hub/journal-policies#imagmanip>. These should be uploaded as 'Supporting information for review process only'. Please label/highlight the original gels so that we can clearly see which sections/lanes have been used in the manuscript figures.

Full western blot images have now been uploaded as “Supporting information for review process only”.

-Please include an Abstract Figure legend text within the main article file.

The abstract figure was labelled as Figure 7 in the manuscript. This has now been changed to “Abstract Figure” for clarity. An example of the Figure is below.

Abstract Figure: Summary of molecular changes within the fetal lung as a result of MitoQ treatment. Changes in expression of pathways are indicated by direction of arrows; no change in expression is represented by purple sideways double-sided arrows. A main effect of MitoQ is represented by a green arrow. An interaction of hypoxia and MitoQ is indicated by yellow arrows. MitoQ treatment increased the expression of Mitochondrial Complex III and ATP Synthase (Complex V) and expression of Surfactant Proteins B & C. There was an interaction of MitoQ with hypoxia resulting in increased expression of sodium transporter SCNN1A and Surfactant Protein D within the fetal lung.

Dear Janna,

Re: JP-RP-2023-284786R3 "MitoQ as an antenatal antioxidant treatment improves markers of lung maturation in healthy and hypoxic pregnancy" by Mitchell C Lock, Kimberley J Botting, Beth J Allison, Youguo Niu, Sage Ford, Michael P. Murphy, Sandra Orgeig, Dino A Giussani, and Janna L Morrison

We are pleased to tell you that your paper has been accepted for publication in The Journal of Physiology.

Authors should note that it is too late at this point to offer corrections prior to proofing. The accepted version will be published online, ahead of the copy edited and typeset version being made available. Major corrections at proof stage, such as changes to figures, will be referred to the Editors for approval before they can be incorporated. Only minor changes, such as to style and consistency, should be made at proof stage. Changes that need to be made after proof stage will usually require a formal correction notice.

Best wishes,

Laura

Professor Laura Bennet
Senior Editor
The Journal of Physiology
<https://jp.msubmit.net>
<http://jp.physoc.org>
The Physiological Society
Hodgkin Huxley House
30 Farringdon Lane
London, EC1R 3AW
UK
<http://www.physoc.org>
<http://journals.physoc.org>

P.S. - You can help your research get the attention it deserves! Check out Wiley's free Promotion Guide for best-practice recommendations for promoting your work at www.wileyauthors.com/eeo/guide. You can learn more about Wiley Editing Services which offers professional video, design, and writing services to create shareable video abstracts, infographics, conference posters, lay summaries, and research news stories for your research at www.wileyauthors.com/eeo/promotion.

IMPORTANT NOTICE ABOUT OPEN ACCESS: To assist authors whose funding agencies mandate public access to published research findings sooner than 12 months after publication, The Journal of Physiology allows authors to pay an Open Access (OA) fee to have their papers made freely available immediately on publication.

You can check if your funder or institution has a Wiley Open Access Account here: <https://authorservices.wiley.com/author-resources/Journal-Authors/licensing-and-open-access/open-access/author-compliance-tool.html>.